# CCRA: Optimizing Vision-Language Consistency via Cross-Layer Regional Attention Alignment

## Abstract

Vision-Language Models (VLMs) face challenges in effectively coordinating diverse cross-attention mechanisms for visual-language alignment, leading to attention drift and suboptimal performance. We propose Consistent Cross-layer Regional Alignment (CCRA), which introduces Layer-Patch-Wise Cross Attention (LPWCA) to capture fine-grained regional-semantic correlations by jointly weighting patch and layer-wise embedding. Also, we employ a novel Progressive Attention Integration (PAI) that systematically coordinates patch-layer-wise, layer-wise, and patch-wise attention mechanisms in sequence. This progressive design ensures consistency from semantic to regional levels while preventing attention drift and maximizing each attention's benefits. Experimental results on eleven diverse vision-language benchmarks demonstrate that our CCRA-enhanced VLMs achieves state-of-the-art performance, outperforming all baseline methods with only 3.55M additional parameters, while providing enhanced interpretability through more regionally-focused and semantically-aligned attention patterns.

## 1 Introduction

Vision-Language Models (VLMs) have fundamentally transformed visual question answering (Jia et al., 2024), object detection (Liu et al., 2024b), segmentation (Khan et al., 2022), OCR (Singh et al., 2019b), etc. A key insight is that diverse tasks, expressed through different text queries, demand very different kinds of information from the same image. This difference concerns not only which regions should be attended to, but also which embedding layers deserve greater emphasis when transferring semantic information (Lin et al., 2025b). This presents a fundamental challenge: how to optimize vision information extraction to better align with the specific needs of text queries for optimal performance.

Existing approaches to vision-language alignment fall into several categories. Some methods extract image embeddings from specific layers of the vision encoder and then perform Patch-Wise Cross Attention (PWCA) between textual and visual embeddings, as in VC-GPT (Luo et al., 2022), Flamingo (Alayrac et al., 2022), and TiMix (Jiang et al., 2024). However, diverse tasks often require a different emphasis on visual features at multiple semantic levels (Wu et al., 2022). To address this limitation in vision–language alignment, other approaches employ Layer-Wise Cross Attention (LWCA) to assign importance weights across different layers, as in IGVA (Li et al., 2025b), MLVF (Lin et al., 2025a), Dense Connector (Yao et al., 2024a) and MMFuser (Cao et al., 2024).

Despite these advances, a critical limitation persists: harmonic coordination between diverse attention mechanisms lacks effective organization, potentially leading to mismatched attention from different perspectives and resulting in suboptimal performance and poor interpretability. To address this limitation, we propose **Consistent Cross-layer Regional Alignment (CCRA)** with two key contributions:

1. **Layer-Patch-Wise Cross Attention (LPWCA)**: Beyond existing LWCA and PWCA, we introduce LPWCA to capture fine-grained regional-semantic correlations, enabling superior performance across diverse tasks.

2. **Progressive Attention Integration (PAI)**: We systematically integrate all three attention mechanisms through progressively operating LPWCA, optimized Gaussian-smoothed LWCA and finally PWCA. This design maximizes the benefits of individual attention mechanisms while ensuring consistency in both semantic and regional levels, enhancing both performance and interpretability.

To demonstrate CCRA's effectiveness in improving generalization performance and interpretability, we evaluate our CCRA-enhanced LLaVA-v1.5-7B model on diverse vision tasks and visualize attention patterns through feature heatmaps. Our results demonstrate that the proposed model outperforms all baseline methods across markedly different tasks with diverse task queries. Meanwhile, the feature heatmaps visualize the adaptivity and consistency of feature attention, which supports the superior performance of VLM across diverse tasks, and also provide more interpretable visual representations of feature importance compared to existing approaches.

## 2 RELATED WORK

### 2.1 VLM WITHOUT VISION-LANGUAGE ALIGNMENT

Conventional VLMs often decouple the processing of visual and textual embeddings, e.g., LLaVA (Liu et al., 2023), MiniGPT-v2 (Chen et al., 2023), and LLaMA-Adapter-v2 (Gao et al., 2023). They often extract a single-layer embedding from the visual encoder and feed it together with textual embeddings to pre-trained encoders such as CLIP (Jiang et al., 2023). However, diverse tasks often require visual features from a broader range of semantic levels (Iana et al., 2024). Accordingly, recent advances leverage cross-layer visual features for comprehensive representations. These approaches capture both low-level details from early layers and high-level semantics from deeper layers. To reduce feature redundancy and noise, these methods also involved similarity-based (Raghu et al., 2021; Yao et al., 2024b; Sun et al., 2025) and proportion-based (Cao et al., 2024; Chen et al., 2024a;b) layer feature selection has been explored. However, these methods operate independently of textual input, failing to consider that different tasks have varying visual requirements. Early layers handle color and many spatial tasks such as counting or localization well (Chen et al., 2025; Yao et al., 2024b). OCR is also sensitive to visual details in the shallow layers, and insufficient low-level information may lead to recognition errors (Cao et al., 2024). By contrast, high-level semantic reasoning, long-horizon action understanding, and knowledge-intensive question answering rely on the deepest visual representations (Li et al., 2025a). Such methods do not consider text-image alignment, leading to suboptimal VLM performance.

### 2.2 VLM WITH VISION-LANGUAGE ALIGNMENT

Recent advances in vision-language alignment have explored various mechanisms to bridge textual semantics with visual representations. One line of research emphasizes PWCA, where image embeddings extracted from specific layers of the vision encoder are aligned with textual queries through cross-attention, as in VC-GPT (Luo et al., 2022), Flamingo (Alayrac et al., 2022), TiMix (Jiang et al., 2024), and EVEv2 (Diao et al., 2025). This approach enhances fine-grained regional control and enriches visual representation, making it particularly effective for tasks requiring precise regional alignment (Yue et al., 2024). Another line focuses on LWCA, which aggregates visual features from multiple encoder layers, often guided by the textual instruction, to adaptively weight semantic levels. Representative works include the Instruction-Guided Vision Aggregator (IGVA) (Li et al., 2025b), while other works such as MLVF (Lin et al., 2025a), the Dense Connector (Yao et al., 2024a), and MMFuser (Cao et al., 2024) further highlight the benefit of leveraging multi-layer visual signals.

Despite their respective strengths, LWCA and PWCA are typically designed independently, often lacking consistency between regional and semantic focus. This decoupled design leads to attention drift, where attention across layers inconsistently shifts regions of focus, undermining stable alignment and interpretability (Li et al., 2025a). Moreover, relying solely on one form of attention neglects the relative importance between regional location and semantic depth, limiting the model's ability to effectively optimize vision-language features.

To address these limitations, recent works have attempted to combine PWCA and LWCA. For example, Liu et al. (2025) proposed a unified framework that compresses patch-level information via

MLPs and integrates it with cross-layer attention, offering a more holistic alignment. However, rigid coordination may lead to inorganic coordination between different attentions and provide suboptimal performance on complex multimodal tasks (Nam et al., 2017; Liu et al., 2025).

# 3 METHODOLOGY

As discussed above, diverse attention mechanisms have their specific benefits, but a mechanism is needed to harmoniously integrate all these visual-language cross attentions to globally optimize VLM's performance across different tasks. In addition, considering the need for human being's understanding, we also need to consider the feature interpretability as a further constraint. To reflect these considerations, we propose **Consistent Cross-layer Regional Alignment (CCRA)**, a novel framework to unify diverse visual-language cross attention under one umbrella for optimal task-oriented performance, and also support consistent feature attention for interpretable understanding. CCRA differs from the previous work in the following two aspects, which is shown in Figure 1.

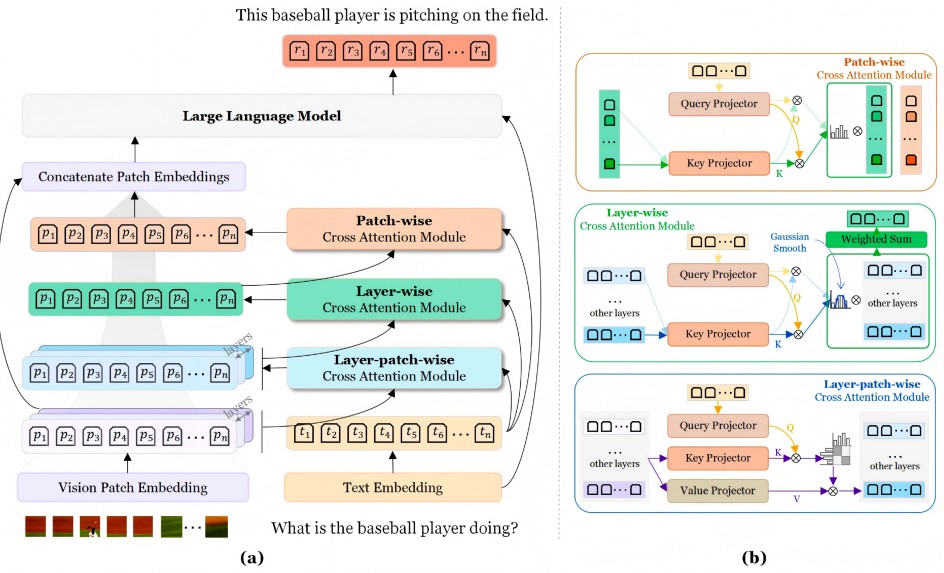

Figure 1: (a) An overview of our VLM with Consistent Cross-layer Regional Alignment. LPWCA, LWCA, and PWCA are progressively used to align both textual embedding and visual embedding gradually for optimal task-oriented performance (b) The detailed illustration of LPWCA, LWCA and PWCA, where LPWCA provides joint and global optimization between regional and semantic information; the optimized Gaussian-smoothed LWCA provides continuous attention along semantic aspect; the PWCA provides the consistency constraint along regional apsect.

1. **Layer-Patch-Wise Cross Attention (LPWCA)**: we first introduced LPWCA, to complement exising LWCA and PWCA. By such a design, we connect the correlation between layer- and patch-wise information, thus provide a finer-grained feature control than merely LWCA and PWCA. (Sec. 3.1)

2. **Progressive Attention Integration (PAI)**: Beyond LPWCA, We proposed PAI to harmoniously unify all three attention mechanisms (LPWCA, LWCA, PWCA) in progressive granularity sequence for optimal task-oriented performance, and provide consistent semantic and regional feature attention for human beings' understanding. (Sec.3.2)

## 3.1 LAYER-PATCH-WISE CROSS ATTENTION

In addition to LWCA and PWCA, which provide semantic and regional attention respectively, we complement them with the layer-patch-wise cross attention (LPWCA) as a fundamental operation, to reflect the global and joint importance on both aspects.

To do so, the multi-layer visual features extracted from a visual encoder (e,g., CLIP ViT (Radford et al., 2021)): $\mathbf{F}_v^l \in \mathbb{R}^{N \times d}$, where $l \in \{1, 2, \ldots, L\}$ is the layer index, and $N$, $d$ are the number of image patches and the feature dimension, respectively, are flattened into a unified patch-layer feature sequence $\mathbf{F}_{\text{stack}}$:

$$\mathbf{F}_{\text{stack}} = \left[ \mathbf{F}_v^1; \mathbf{F}_v^2; \cdots ; \mathbf{F}_v^L \right] \in \mathbb{R}^{L \times (N \times d)}. \tag{1}$$

Through such a way, the hierarchical structure of feature space, which can be viewed from patch and layer perspective, are unified into the same space.

Then in order to align with the textual query, we first process the query into a set of textual embeddings $\mathbf{F}_t \in \mathbb{R}^{T \times d}$. A self-attention module is first applied over $\mathbf{F}_t$ to compute token-contextualised importance scores $\boldsymbol{\alpha}_t \in \mathbb{R}^T$, which indicate the relative contribution of each token in guiding the visual alignment.

$$\boldsymbol{\alpha}_t = \text{Softmax}(\text{SelfAttention}(\mathbf{F}_t)). \tag{2}$$

Next, the textual embeddings $\mathbf{F}_t$ are projected into a query space $\mathbf{Q}(\mathbf{F}_t)$, while the stacked visual features $\mathbf{F}_{\text{stack}}$ are projected into a key space $\mathbf{K}(\mathbf{F}_{\text{stack}})$. The layer-patch attention scores $\mathbf{A}_{lp} \in \mathbb{R}^{T \times (L \times N)}$ can thus be computed:

$$\mathbf{A}_{lp} = \frac{1}{\sqrt{d}} \mathbf{Q}(\mathbf{F}_t) \mathbf{K}(\mathbf{F}_{\text{stack}})^\top. \tag{3}$$

Then $\mathbf{A}_{lp}$ is aggregated across all textual tokens using the learned importance weights $\boldsymbol{\alpha}_t$ to form a unified attention map $\mathbf{W}_{lp} \in \mathbb{R}^{L \times N}$ over spatial patches and layers:

$$\mathbf{W}_{lp} = \boldsymbol{\alpha}_t^\top \mathbf{A}_{lp}, \tag{4}$$

Such a map demonstrates the global importance of every patch feature, regardless of where and which layer it is located at. This map is then used to modulate the original stacked features $\mathbf{F}_{\text{stack}}$ via element-wise multiplication, followed by a residual connection and layer normalization $LN(\cdot)$:

$$\mathbf{F}_{\text{lp}} = \text{LN}(\mathbf{F}_{\text{stack}} \odot \mathbf{W}_{lp} + \mathbf{F}_{\text{stack}}). \tag{5}$$

where $\mathbf{F}_{\text{lp}} \in \mathbb{R}^{L \times (N \times d)}$ are the features aligned with the textual query from a joint patch-layer perspective. These features are then reshaped back to $\mathbb{R}^{L \times N \times d}$ to recover the per-layer structure for the next stage. With such an attention mechanism, we provide a more comprehensive textual-image alignment than only considering the patch- or layer-wise impact.

### 3.2 Progressive Attention Integration

Although LPWCA provides finer-grained attention, LWCA is crucial for focusing on semantically relevant layers, while PWCA constrains attention to consistent regions across layers. Without them, the learned features could be semantically or spatially inconsistent, making them difficult for humans to interpret. Therefore, Progressive Attention Integration (PAI) is proposed to integrate all three mechanisms harmoniously.

**Integration with LWCA.** Based on the globally-aligned features $\mathbf{F}_{\text{lp}}$ from LPWCA (reshaped to $L \times N \times d$), we apply a revised LWCA to provide continuous semantic attention. Specifically, the visual features are first spatially averaged to obtain a set of layer-level descriptors:

$$\mathbf{F}_{\text{layer}} = \left[ \text{AvgPool}(\mathbf{F}_{\text{lp}}^1); \cdots ; \text{AvgPool}(\mathbf{F}_{\text{lp}}^L) \right] \in \mathbb{R}^{L \times d}. \tag{6}$$

Then, cross-attention scores $\mathbf{A}_l \in \mathbb{R}^{T \times L}$ are computed between textual embeddings and layer descriptors, followed by aggregation using the same token importance weights $\boldsymbol{\alpha}_t$:

$$\mathbf{A}_l = \frac{1}{\sqrt{d}} \mathbf{Q}(\mathbf{F}_t) \mathbf{K}(\mathbf{F}_{\text{layer}})^\top; \tag{7}$$

$$\mathbf{w}_l = \boldsymbol{\alpha}_t^\top \mathbf{A}_l, \quad \mathbf{w}_l \in \mathbb{R}^L. \tag{8}$$

Previous approaches to LWCA often select specific layers or cluster them to avoid sharp, noisy transitions in attention weights, which could disrupt the semantic smoothness across layers (Sung et al., 2023; Li et al., 2025b; Lin et al., 2025c). However, this strategy risks discarding valuable information from the omitted layers. To address this, we introduce a Gaussian smoothing kernel applied to the raw layer attention scores $\mathbf{w}_l$. This method allows us to utilize information from all layers while simultaneously enforcing a smooth attention distribution, thus obtaining the final, refined layer weights $\hat{\mathbf{w}}_l \in \mathbb{R}^L$ that maintain both completeness of information and semantic consistency.

The semantically-aligned visual representation $\mathbf{F}_{\text{semantic}} \in \mathbb{R}^{N \times d}$ is derived via a weighted aggregation of the globally-aligned layer features. Let $\hat{\mathbf{F}}_{\text{lp}}$ be the result of the weighted sum. A residual connection and layer normalization are then applied:

$$\hat{\mathbf{F}}_{\text{lp}} = \sum_{l=1}^{L} \hat{w}_{l,l} \cdot \mathbf{F}_{\text{lp}}^{l} \tag{9}$$

$$\mathbf{F}_{\text{semantic}} = \text{LN}\left(\hat{\mathbf{F}}_{\text{lp}} + \text{AvgPool}(\hat{\mathbf{F}}_{\text{lp}})\right). \tag{10}$$

**Integration with PWCA.** Furthermore, to maintain regional consistency, we apply PWCA on $\mathbf{F}_{\text{semantic}}$. We first compute cross-attention between language tokens and the patch features of $\mathbf{F}_{\text{semantic}}$:

$$\mathbf{A}_p = \frac{1}{\sqrt{d}}\mathbf{Q}(\mathbf{F}_t)\mathbf{K}(\mathbf{F}_{\text{semantic}})^{\top}, \tag{11}$$

where $\mathbf{A}_p \in \mathbb{R}^{T \times N}$. The scores are then aggregated using the token importance weights $\boldsymbol{\alpha}_t$ to get patch weights $\mathbf{w}_p \in \mathbb{R}^N$:

$$\mathbf{w}_p = \boldsymbol{\alpha}_t^{\top} \mathbf{A}_p. \tag{12}$$

Finally, a residual connection and layer normalization are applied to obtain the regionally-aligned visual representation $\mathbf{F}_{\text{regional}} \in \mathbb{R}^{N \times d}$:

$$\mathbf{F}_{\text{regional}} = \text{LN}\left(\mathbf{F}_{\text{semantic}} \odot (1 + \mathbf{w}_p)\right). \tag{13}$$

To preserve both the original high-level visual semantics and the newly refined features, we concatenate $\mathbf{F}_{\text{regional}}$ with the original final-layer visual feature $\mathbf{F}_v^L$:

$$\mathbf{F}_{\text{fused}} = [\mathbf{F}_{\text{regional}}; \mathbf{F}_v^L] \in \mathbb{R}^{N \times 2d}. \tag{14}$$

**Visual-textual Feature Fusion** To align with the hidden dimension $d$ of the large language model, we apply a visual projection head $\text{Proj}_{\text{vis}} : \mathbb{R}^{2d} \to \mathbb{R}^d$ to each fused patch token. Subsequently, the resulting visual representation is then concatenated with the textual embeddings $\mathbf{F}_t$ and passed into a large language model for visual-language predictions (e.g., answer generation, captioning):

$$\hat{Y} = \text{LLM}([\text{Proj}_{\text{vis}}(\mathbf{F}_{\text{fused}}); \mathbf{F}_t]). \tag{15}$$

Through such a progressive integration of LPWCA, LWCA, and PWCA, the final visual feature $\mathbf{F}_{\text{fused}}$ from PAI is tightly aligned with textual query, which supports the optimal performacne of VLM after the visual-textual feature fusion. Meanwhile, it is also further constrained in semantic smoothness and regional consistency, which provides understandable attention map for human being.

The effectiveness of CCRA also depends on a few interpretable hyperparameters, such as the layer smoothing kernel size and embedding dimensions. Their impacts are discussed in Appendix C.2. The overall training and inference procedure of CCRA is summarized in Algorithm 1.

---

**Algorithm 1:** Training and Inference of CCRA-based Vision-Language Model

---

**Input:** Image $I \in \mathbb{R}^{H \times W \times 3}$, Text sequence $\mathcal{T}$ with token length $T$, Task label $Y$ (for training)
**Output:** Prediction $\hat{Y}$ or updated model parameters

**1. Visual and Text Encoding**
$\mathbf{F}_{\text{stack}} \leftarrow \text{VisualEncoder}(I)$ $\mathbf{F}_t \leftarrow \text{TextEncoder}(\mathcal{T})$

**2. Consistent Cross-layer Regional Alignment (CCRA)**
$\mathbf{F}_{\text{lp}} \leftarrow \text{LPWCA}(\mathbf{F}_t, \mathbf{F}_{\text{stack}})$ $\mathbf{F}_{\text{semantic}} \leftarrow \text{LWCA}(\mathbf{F}_t, \mathbf{F}_{\text{lp}})$ $\mathbf{F}_{\text{regional}} \leftarrow \text{PWCA}(\mathbf{F}_t, \mathbf{F}_{\text{semantic}})$ $\mathbf{F}_{\text{fused}} \leftarrow \text{Fuse}(\mathbf{F}_{\text{regional}}, \mathbf{F}_v^L)$

**3. Training Stage 1: Feature Alignment Pretraining**
$\hat{Y} \leftarrow \text{LLM}([\text{Proj}_{\text{vis}}(\mathbf{F}_{\text{fused}}); \mathbf{F}_t])$ ;          `// Caption prediction; freeze VisualEncoder and LLM`
$\mathcal{L}_{\text{pretrain}} \leftarrow \text{CrossEntropy}(\hat{Y}, Y)$

**4. Training Stage 2: End-to-End Finetuning**
$\hat{Y} \leftarrow \text{LLM}([\text{Proj}_{\text{vis}}(\mathbf{F}_{\text{fused}}); \mathbf{F}_t])$ ;          `// Task-specific prediction; freeze VisualEncoder only`
$\mathcal{L}_{\text{finetune}} \leftarrow \text{CrossEntropy}(\hat{Y}, Y)$

**5. Model Inference (if label $Y$ is not available)**
**Execute steps 1 and 2, then skip steps 3 and 4**
$\hat{Y} \leftarrow \text{LLM}([\text{Proj}_{\text{vis}}(\mathbf{F}_{\text{fused}}); \mathbf{F}_t])$

---

## 4 EXPERIMENT

We first outline the experimental setup (Section 4.1). We then evaluate CCRA on eleven public benchmarks spanning compositional reasoning, OCR, instruction following, and domain-specific tasks, comparing against state-of-the-art baselines (Table 1). Beyond accuracy, we analyze the attention behavior of LPWCA/LWCA/PWCA with quantitative consistency metrics and qualitative heatmaps (Section 4.2), compare coordination strategies in an integration study (Section 4.3), and validate each component via ablations (Section 4.4).

### 4.1 EXPERIMENTAL SETUP

We follow the two-stage training strategy of LLaVA-v1.5-7B. In the pre-training stage on LLaVA-LCS-558K, CCRA is not applied due to the lack of annotations. In the instruction-tuning stage on LLaVA-Instruct-665K, CCRA is integrated to enhance vision–language consistency. Apart from this modification, the optimization procedure is identical to LLaVA. Further details are provided in Appendix B.

We evaluate CCRA on eleven widely-used benchmarks with diverse task requirements, they are GQA (Hudson & Manning, 2019), ScienceQA(SQA) (Lu et al., 2022), TextVQA (Singh et al., 2019a), VizWiz (Gurari et al., 2018), MMB-en (Liu et al., 2024c), MM-Vet (Yu et al., 2023), SEED-I (Li et al., 2024), MMMU (Yue et al., 2024), MME-p (Fu et al., 2023a), and POPE (Li et al., 2023b), Notably, for ScienceQA, we only evaluate on the set with image context. More details on dataset can be found in Appendix A. Parallelly, SOTA methods including LLaVA-v1.5-7B (Liu et al., 2024a), LLaVA-v1.5-13B (Liu et al., 2024a), mPLUG-Owl2 (Ye et al., 2024), MiniGPT-v2 (Chen et al., 2023), LLaMA-Adapter-v2 (Gao et al., 2023), IDEFICS (Laurençon et al., 2023), Flamingo (Alayrac et al., 2022), DenseConnector (Yao et al., 2024b), MMFuser (Cao et al., 2024), IGVA (Li et al., 2025a) and Qwen-VL-Chat (Bai et al., 2023), are used to compare with CCRA, to demonstrate CCRA's advance.

### 4.2 RESULTS AND ANALYSIS

#### 4.2.1 OVERALL PERFORMANCE

CCRA achieves the best results across eleven benchmarks in Table 1. Accuracy is used for most tasks, MME-p uses the official MME-p score and POPE uses the F1 score. The additional attention operations introduce only 3.55M parameters, which is negligible relative to a seven billion parameter backbone, yet they allow LLaVA-v1.5-7B with CCRA to surpass LLaVA-v1.5-13B. This shows that even lightweight consistency constraints can yield substantial improvements in vision-language alignment.

Table 1: Comparison across 11 benchmarks. Models are grouped by whether they adopt vision-language alignment.

| Model | LLM | Resolution | Train Data | GQA | SQA | TextVQA | VizWiz | MMB-en | MM-Vet | SEED-I | MMMU | MME-p | POPE |
|---|---|---|---|---|---|---|---|---|---|---|---|---|---|
| Metric | | | | Acc(%) | Acc(%) | Acc(%) | Acc(%) | Acc(%) | Acc(%) | Acc(%) | Acc(%) | MME-p Score | F1-Score |
| **Models without Vision-Language Alignment** | | | | | | | | | | | | | |
| LLaVA-v1.5-7B | Vicuna-v1.5-7B | 336 | 0.5M+0.6M | 61.9 | 67.1 | 58.1 | 53.2 | 63.9 | 32.8 | 67.2 | 34.9 | 1480.6 | 86.9 |
| LLaVA-v1.5-13B | Vicuna-v1.5-13B | 336 | 0.5M+0.6M | 63.3 | 71.0 | 61.3 | 53.6 | 67.7 | 36.1 | 68.2 | 34.9 | - | 87.2 |
| MiniGPT-v2 | LLaMA 2-7B | 448 | - | 60.1 | - | - | 53.6 | 9.4 | - | - | - | - | - |
| IDEFICS | LLaMA-7B | 224 | 1.6B | 38.4 | - | 25.9 | 35.5 | 48.2 | - | - | - | - | - |
| LLaMA-Adapter-v2 | LLaMA-7B | 336 | 0.6M | - | - | - | - | 41.0 | 31.5 | 32.7 | 29.8 | 972.7 | - |
| mPLUG-Owl2 | LLaMA 2-7B | 448 | 384M+1.2M | 56.1 | 68.7 | 54.3 | 54.5 | 64.5 | 36.2 | 57.8 | - | 1450.2 | 86.2 |
| Qwen-VL-Chat | Qwen-7B | 448 | 1.4B+50M+0.3M | 57.5 | 68.2 | 61.5 | 38.9 | 60.6 | - | 65.4 | 35.9 | 1487.6 | - |
| **Models with PWCA-Enhanced Vision-Language Alignment** | | | | | | | | | | | | | |
| Flamingo | Chinchilla-70B | 336 | 43M + 185M | - | - | 37.9 | 49.8 | - | - | - | - | - | - |
| **Models with LWCA-Enhanced Vision-Language Alignment** | | | | | | | | | | | | | |
| DenseConnector | - | - | - | 63.8 | 69.5 | 59.2 | - | 66.8 | 32.7 | - | 34.8 | - | 86.6 |
| MMFuser | Vicuna-13B | 336 | 0.5M+0.6M | 62.8 | 68.7 | 58.8 | 53.4 | 67.5 | 32.6 | 60.8 | - | 1479.7 | 86.3 |
| IGVA | Vicuna-v1.5-7B | 336 | 0.5M+0.6M | 63.1 | 70.2 | 59.4 | 54.3 | 66.9 | 33.5 | 68.3 | 36.4 | 1519.8 | 87.8 |
| **Model with CCRA-Enhanced Vision-Language Alignment** | | | | | | | | | | | | | |
| **Ours** | Vicuna-v1.5-7B | 336 | 0.5M+0.6M | **64.2** | **71.3** | **63.1** | **54.6** | **67.9** | **37.5** | **69.6** | **37.6** | **1525.6** | **88.9** |

**Performance Improvements.** A closer look at Table 1 highlights where the improvements come from. CCRA surpasses IGVA by 1.1 on GQA, 1.3 on SEED-I, and 1.0 on MMB-en, and exceeds

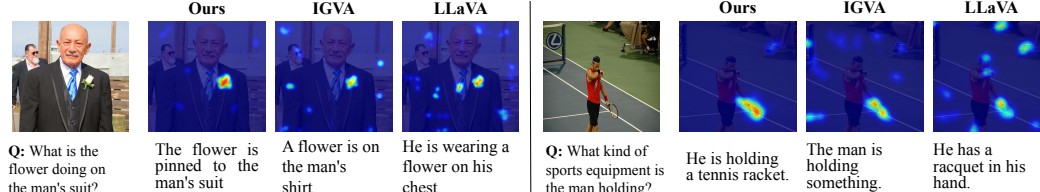

Figure 2: Cross-attention maps averaged over mid-layers of the LLM. CCRA directs attention more consistently to task relevant regions compared to IGVA and LLaVA baselines, resulting in answers that align better with visual evidence.

MMFuser by 4.3 on TextVQA and 4.9 on MM-Vet. These improvements arise from the LPWCA and PAI modules as shown in Table 4 and Table 3, which refine alignment through layer–patch correlation and progressive integration of attention mechanisms.

**Interpretability Improvements.** We further examine how CCRA improves the use of visual features inside the language model. Figure 2 shows that CCRA produces features that guide the LLM to focus on the relevant regions of the task, producing accurate and semantically aligned responses. In contrast, IGVA and LLaVA lead to more diffuse and inconsistent attention, often associated with vague or incorrect output.

### 4.2.2 ATTENTION ANALYSIS

**Layer-Patch-Wise Cross Attention Behavior.** LPWCA aligns spatial regions across layers, yielding more consistent focus. Quantitatively, it reduces cross-layer divergence (lower JS), improves similarity (higher cosine), stabilizes spatial centroids (lower drift), and increases patch stability. In addition, LPWCA shows stronger agreement with the final PWCA map and induces smoother LWCA weights (lower TV, higher entropy), indicating more coherent integration across modules. All metrics are summarized in Table 2 and formally defined in Appendix C.1, where they are computed on both pre- and post-LPWCA features under identical projections and normalization. Qualitatively, Figure 3 visualizes patch–layer scores from representative layers (12, 18, 24), showing that LPWCA consistently highlights shoes and shirts across depth, confirming stable region–layer alignment under language guidance.

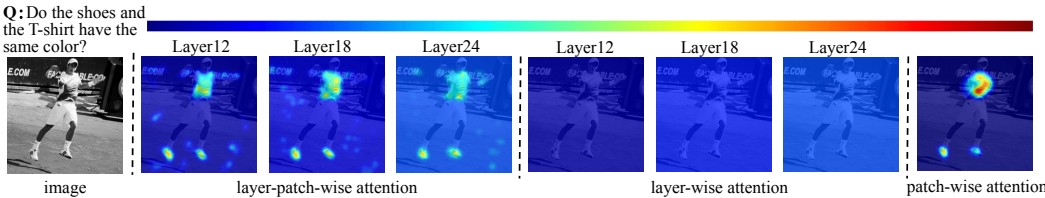

Figure 3: Qualitative visualization of the three attention modules in CCRA under the query *"Do the shoes and the T-shirt have the same color?"*. From left to right: input image and question, LPWCA, LWCA and PWCA. Together, these modules produce consistent semantic-regional alignment.

Table 2: Attention consistency metrics before and after LPWCA ($L$=24). All metrics are computed on $F_{stack}$ (pre-LPWCA) and $F_{lp}$ (post-LPWCA) using identical projections and normalization.

| Metric | JS-avg ↓ | Cos-avg ↑ | Drift ↓ | Std$_{patch}$ ↓ | JS-to-final ↓ | Cos-to-final ↑ | TV($w$) ↓ | Entropy $H(w)$ ↑ |
|---|---|---|---|---|---|---|---|---|
| $F_{stack}$ | 0.218 | 0.731 | 0.064 | 0.052 | 0.173 | 0.754 | 0.060 | 2.41 |
| $F_{lp}$ | **0.147** | **0.812** | **0.041** | **0.038** | **0.119** | **0.835** | **0.028** | **2.95** |

**Layer-Wise Cross Attention Behavior.** LWCA identifies the semantic depth most relevant to a query. As shown in Figure 4, appearance questions activate lower and middle layers, while reasoning questions shift focus to deeper layers, and removing Gaussian smoothing leads to performance drops

(Table 4). For qualitative analysis, we visualize layer attention scores across depth after smoothing and select representative layers (12, 18, 24), mapping them back to patches. As shown in Figure 3, LWCA emphasizes semantic depth while leaving spatial grounding to PWCA.

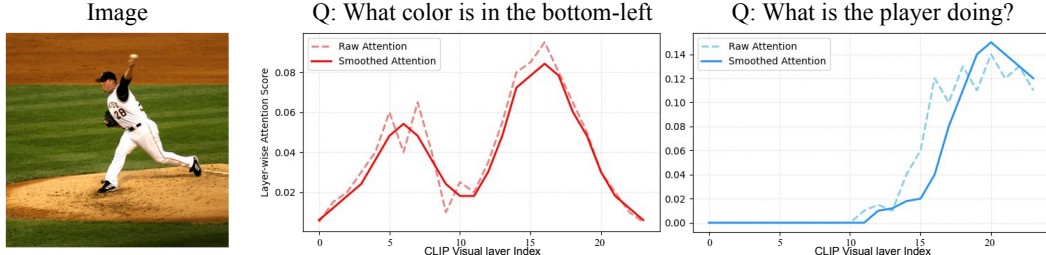

Figure 4: Comparison of LWCA distributions for queries of different semantic levels. Shallow appearance-based queries activate earlier layers, while high-level reasoning queries attend to deeper layers. Smoothed attention curves (solid) reveal more coherent trends than raw attention (dashed).

**Patch-Wise Cross Attention Behavior.** PWCA provides fine-grained spatial grounding on the fused representation. Removing PWCA leads to clear drops on TextVQA, VizWiz, and MMMU (Table 4), confirming its importance for OCR- and grounding-heavy tasks. For qualitative analysis, we visualize the attention scores of PWCA as shown in Figure 3, PWCA sharply localizes the shoes and shirt, complementing LPWCA's region–layer alignment and LWCA's semantic weighting.

## 4.3 ATTENTION INTEGRATION STUDY

To better prove why PAI is superior to other integration strategies, we compared **PAI** (Figure 5 (c)) with two of its variants (Figure 5): (a) **Decoupled Integration**, where patch-wise and layer-wise cross attentions are fused in parallel before entering the LLM; and (b) **Shuffled Integration**, which reverses the order of patch- and layer-wise operations in PAI.

The performance is reported in Table 3, where our proposed PAI demonstrates the best performance. Furthermore, the text-attention is visualized in Figure 6 to intuitively check the vision-language alignment. The figure shows that the PAI produces sharper, more consistent, and semantically focused attention, while both variants exhibit dispersed or misaligned patterns. These results prove the effectiveness of our dedicated design for PAI.

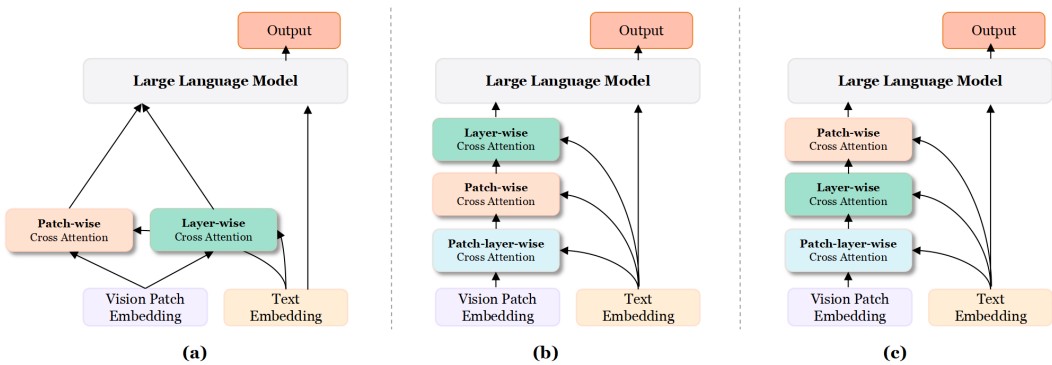

Figure 5: Comparison of cross-attention coordination strategies for combining patch-level (regional) and layer-level (semantic) information. **(a) Decoupled Integration**: PWCA and LWCA attentions are applied independently and fused in parallel before LLM input. **(b) Shuffled Integration**: A reversed ordering of patch-wise and layer-wise attention compared to PAI. **(c) Progressive Attention Integration**: Our progressive strategy that sequentially applies patch-layer-wise, layer-wise, and patch-wise attention to iteratively refine visual features.

Table 3: Performance comparison between PAI and its variants

| Model Variant | GQA | SQA | TextVQA | VizWiz | MMB-en | MM-Vet | SEED-I | MMMU | MME-p | POPE |
|---|---|---|---|---|---|---|---|---|---|---|
| Decoupled Integration | 63.1 | 70.2 | 58.7 | 53.0 | 67.0 | 36.5 | 68.5 | 36.6 | 1497.7 | 87.4 |
| Shuffled Integration | 63.8 | 70.8 | 59.3 | 53.7 | 67.3 | 37.0 | 68.9 | 36.8 | 1501.6 | 88.2 |
| **Ours (PAI)** | **64.2** | **71.3** | **63.1** | **54.6** | **67.9** | **37.5** | **69.6** | **37.6** | **1525.6** | **88.9** |

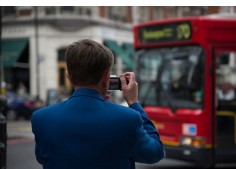 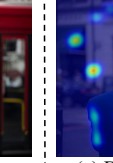 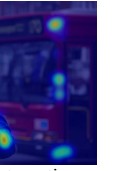 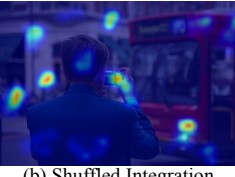 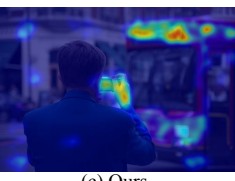

| Question: What is the man doing? | (a) Decoupled integration | (b) Shuffled Integration | (c) Ours |
|---|---|---|---|

Figure 6: Cross-attention maps from the LLM under different coordination strategies. CCRA (c) yields sharper and more semantically aligned attention compared to (a) independent coordination and (b) a variant progressive design. These visualizations confirm the effectiveness of our full progressive attention refinement.

## 4.4 ABLATION STUDY

We conduct ablations on each attention mechanism (Table 4). PWCA mainly benefits fine-grained tasks (TextVQA, VizWiz, MMMU), LWCA supports semantic reasoning (SQA, MMB-en, SEED-I), and LPWCA ensures cross-layer consistency (MM-Vet, POPE). Gaussian smoothing in LWCA yields small but consistent gains (e.g., –0.4 on SQA, –0.7 on POPE) and stabilizes attention distributions, as illustrated in Figure 4.

Table 4: Ablation study results of CCRA

| Model Variant | GQA | SQA | TextVQA | VizWiz | MMB-en | MM-Vet | SEED-I | MMMU | MME-p | POPE |
|---|---|---|---|---|---|---|---|---|---|---|
| **Ours (CCRA)** | **64.2** | **71.3** | **63.1** | **54.6** | **67.9** | **37.5** | **69.6** | **37.6** | **1525.6** | **88.9** |
| w/o PWCA | 62.9 | 69.5 | 58.8 | 53.2 | 67.1 | 36.8 | 68.3 | 36.4 | 1513.2 | 87.5 |
| w/o LWCA | 63.4 | 70.1 | 58.2 | 53.9 | 66.7 | 36.0 | 68.1 | 36.9 | 1510.3 | 87.0 |
| w/o LPWCA | 62.7 | 70.6 | 59.0 | 52.8 | 66.9 | 37.1 | 68.7 | 36.3 | 1514.6 | 88.7 |
| w/o Gaussian smoothing | 63.8 | 70.9 | 59.1 | 54.0 | 67.3 | 36.5 | 69.0 | 36.7 | 1516.2 | 88.2 |

## 5 CONCLUSION

We proposed Consistent Cross-layer Regional Alignment (CCRA), which integrates layer-patch and progressive attention for more stable visual-language alignment. Layer-Patch-wise Cross Attention (LPWCA) captures fine-grained regional-semantic correlations by jointly considering spatial and depth-wise semantics, providing more comprehensive visual-language alignment than existing patch-wise or layer-wise mechanisms alone. Progressive Attention Integration (PAI) systematically coordinates all three attention mechanisms in sequence, ensuring consistency from semantic to regional levels while maximizing eac attention benefits and preventing attention drift. Experimental results across eleven benchmarks demonstrate that our CCRA-enhanced model achieves state-of-the-art performance, outperforming all baseline methods while adding only 3.55M parameters, with visualization analyses confirming more focused and semantically aligned attention patterns.

Beyond the advanced performance of CCRA, we realize that there is still an optimization space to be explored. Future work could explore alternative smoothing mechanisms for optimizing LWCA beyond the current Gaussian smoothing approach to achieve better integration within the progressive attention framework and further improve vision-language alignment performance.

ETHICS STATEMENT

This work proposes CCRA, a framework for optimizing cross-layer and regional attention consistency in vision-language models. All experiments were conducted using publicly available datasets and benchmarks in the multimodal community, such as GQA, ScienceQA, TextVQA, VizWiz, MM-Bench, MM-Vet, SEED-I, MMMU, MME-p, and POPE. These datasets are widely adopted for academic research and do not contain sensitive personal information. No human subjects, private data, or animal studies were involved.

The proposed method is developed purely for advancing multimodal alignment research and poses minimal direct ethical concerns. However, as with any powerful vision-language system, potential misuse remains possible in domains such as surveillance, disinformation, or decision-making in safety-critical applications. To reduce risks, we emphasize that this work is intended solely for academic research. Responsible use requires careful evaluation of societal impacts, robustness against hallucinations, and deployment only under appropriate safeguards.

REPRODUCIBILITY STATEMENT

To ensure reproducibility, we will release the full implementation of CCRA, including model code, training pipelines, and inference scripts. All datasets employed in this study are publicly accessible. The paper and appendix provide complete details of model architecture, hyperparameters, optimization schedules, and computational resources. Our training follows the established two-stage LLaVA pipeline, with CCRA integrated as an additional module.

We further report extensive ablations, hyperparameter sensitivity analyses, and qualitative visualizations of attention maps to validate each component of the framework. Upon acceptance, we will make pretrained checkpoints, dataset preparation instructions, and experimental logs available to support transparent verification and future research.

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

## A DATASET OVERVIEW

Detailed information about all our 10 datasets is introduced as follows.

**GQA** GQA (Hudson & Manning, 2019) is a large-scale benchmark for compositional visual reasoning, comprising real-world images annotated with structured scene graphs. It contains over 22 million questions across 100K images, designed to evaluate multi-step reasoning that involves attributes, spatial relations, and logical consistency.

**SQA** The Spatial Question Answering (SQA) (Hu et al., 2019) dataset evaluates a model's ability to comprehend spatial relationships between objects. It features questions derived from the VQA dataset, with a specific focus on prepositions such as "left of" or "above."

**TextVQA** TextVQA (Singh et al., 2019b) evaluates the ability to answer questions that require recognizing and interpreting text within images. The dataset includes over 45K questions on images with naturally occurring text, requiring robust OCR and multimodal reasoning capabilities.

**VizWiz** VizWiz (Gurari et al., 2018) contains visual questions posed by visually impaired users. The dataset consists of real-world, often noisy images and naturally phrased questions, establishing it as a benchmark for accessibility-focused VQA and open-ended question understanding.

**MMBench-en** MMBench-en (Li et al., 2023a) is a comprehensive benchmark that evaluates English-language multimodal models across multiple tasks. It features classification-style questions covering perception, reasoning, and instruction-following to assess model generalization across diverse domains.

**MM-Vet** MM-Vet (Fu et al., 2023b) is a curated benchmark that probes vision-language alignment and reasoning consistency. It includes challenging visual questions with distractors, evaluating a model's ability to perceive images accurately rather than relying on language priors. MM-Vet employs a normalized score, computed by averaging accuracies across seven core visual reasoning skills, each weighted equally.

**SEED-I** SEED-I (Zhu et al., 2023), a subset of SEED-Bench, evaluates multimodal instruction-following in vision-centric tasks. It includes open-ended prompts that require image understanding, captioning, and grounding based on natural user instructions.

**MMMU** The Massive Multi-discipline Multimodal Understanding (MMMU) benchmark (Zeng et al., 2023) contains over 10K expert-level questions spanning 57 disciplines, such as medicine, law, and physics. It assesses the capacity of models to handle college-level, multi-disciplinary questions.

**MME-p** MME-p (Fu et al., 2023a), the perception-oriented subset of the MME benchmark, evaluates fine-grained visual capabilities, including object counting, attribute recognition, OCR, and positional reasoning. It is designed to isolate core perceptual skills from higher-level cognitive reasoning.

**POPE** The POPE benchmark (Yao et al., 2023) evaluates whether models ground their answers in visual input or rely on spurious language biases. It presents contrastive image-question pairs to assess the strength of visual grounding and robustness against hallucination.

# B   DETAILED EXPERIMENTAL SETUP

In this appendix, we provide a detailed overview of our two-stage training pipeline, covering both optimization settings and resource utilization.

**Training Configurations.**   The complete hyperparameters and hardware configurations for both training stages are summarized in Table 5. In the pre-training stage, we train the model on the LLaVA-LCS-558K dataset for one epoch (2,179 steps), using a batch size of 256 and a learning rate of $1 \times 10^{-3}$, consistent with the LLaVA-v1.5-7B training strategy. For the visual instruction tuning stage, we switch to the LLaVA-Instruct-665K dataset, reduce the batch size to 128, and decrease the learning rate to $1 \times 10^{-5}$ to ensure training stability. Both stages employ a cosine learning rate schedule with a 3% warm-up period and the AdamW optimizer. We conduct distributed training on 8 NVIDIA A100 GPUs (80GB each), with DeepSpeed ZeRO-2 optimization enabled for memory efficiency.

**Resource Usage.**   As detailed in Table 5, the pre-training stage is memory-efficient, as only a small adapter module is trained while the remainder of the model remains frozen. In contrast, the instruction tuning stage involves optimizing the full model, including the language model, vision encoder, and our proposed CCRA module. This full-model optimization leads to an increase in peak GPU memory usage, despite a reduction in batch size. The CCRA module contributes minimally to this increase, adding less than 3GB of memory overhead. Even with 32-bit precision, the entire model can be accommodated on a single A100 80GB GPU. With the exception of the CCRA integration, the training pipeline strictly adheres to the original LLaVA setup to ensure a fair and controlled comparison.

Table 5: Training configuration and resource usage per stage.

| Setting | Pretraining | Visual Instruction Tuning |
|---|---|---|
| Dataset | LLaVA-LCS-558K | LLaVA-Instruct-665K |
| GPUs | 8×A100 80GB | 8×A100 80GB |
| Batch size | 256 | 128 |
| Total steps | 2,179 | 5,194 |
| Epochs | 1 | 1 |
| Learning rate | 1e-3 | 1e-5 |
| Schedule | cosine | cosine |
| Warm-up | 3% | 3% |
| Optimizer | AdamW | AdamW |
| ZeRO Stage | ZeRO-2 | ZeRO-2 |
| Wall-clock time (h) | 4 | 10 |
| Aggregate GPU-hours | 32 | 80 |
| Peak VRAM / card | 73GB | 76GB |

# C ADDITIONAL EXPERIMENTAL RESULTS

## C.1 DEFINITIONS OF ATTENTION CONSISTENCY METRICS

In the main paper (Table 2), we report several metrics to quantify consistency before and after LPWCA. Here we provide precise definitions with complete notation.

Let $F_{\text{stack}} \in \mathbb{R}^{L \times P \times d}$ be stacked encoder features across $L$ layers with $P$ patches per layer, and $F_{\text{lp}} \in \mathbb{R}^{L \times P \times d}$ the features after LPWCA modulation. We use $X \in F_{\text{stack}}, F_{\text{lp}}$ to denote either feature set. Each patch feature $X_{l,p} \in \mathbb{R}^d$ corresponds to patch $p$ at layer $l$.

**Patch scores and per-layer distributions.** We denote by $Q, K : \mathbb{R}^d \to \mathbb{R}^{d_h}$ fixed linear projections for queries and keys. Let $F_q \in \mathbb{R}^d$ be a text-derived query embedding (e.g., from the final text token). The unnormalized score for patch $p$ at layer $l$ is

$$s_l(p; X) = \frac{1}{\sqrt{d_h}} \left\langle Q(F_q), K(X_{l,p}) \right\rangle$$

We normalize within each layer to obtain a patch distribution

$$p_l(p; X) = \frac{\exp(s_l(p; X))}{\sum_{p'=1}^{P} \exp(s_l(p'; X))}.$$

**1. Layer-wise similarity.** We measure the agreement of per-layer patch distributions across all layer pairs:

$$\text{JS-avg}(X) = \tfrac{2}{L(L-1)} \sum_{l<k} \text{JS}\big(p_l(\cdot; X), p_k(\cdot; X)\big).$$

$$\text{Cos-avg}(X) = \tfrac{2}{L(L-1)} \sum_{l<k} \frac{\langle p_l(\cdot; X), p_k(\cdot; X) \rangle}{\|p_l(\cdot; X)\|_2 \, \|p_k(\cdot; X)\|_2}.$$

**2. Spatial drift.** Let $\mathbf{u}_p \in \mathbb{R}^2$ denote the spatial coordinate of patch $p$ on the image grid. The centroid of layer $l$ is

$$\mu_l(X) = \sum_{p=1}^{P} p_l(p; X) \, \mathbf{u}_p$$

**3. Patch-wise stability.** We measure the variance of attention assigned to the same patch across layers:

$$\text{Std}_{\text{patch}}(X) = \tfrac{1}{P} \sum_{p=1}^{P} \text{Std}\big(\{p_l(p; X)\}_{l=1}^{L}\big)$$

**4. Agreement with final PWCA.** Let $p^\star \in \mathbb{R}^P$ be the final PWCA patch distribution produced by the model. We measure its agreement with per-layer distributions:

$$\text{JS-to-final}(X) = \tfrac{1}{L} \sum_{l=1}^{L} \text{JS}\big(p_l(\cdot; X), p^\star\big).$$

$$\text{Cos-to-final}(X) = \tfrac{1}{L} \sum_{l=1}^{L} \frac{\langle p_l(\cdot; X), p^\star \rangle}{\|p_l(\cdot; X)\|_2 \, \|p^\star\|_2}.$$

**5. Layer weight smoothness (LWCA only).** Let $w \in \mathbb{R}^L$ denote the normalized weights assigned by LWCA across layers. We report their total variation and entropy:

$$\text{TV}(w) = \tfrac{1}{L-1} \sum_{l=1}^{L-1} |w_{l+1} - w_l|, \qquad H(w) = - \sum_{l=1}^{L} w_l \log w_l$$

All metrics are computed for both $F_{\text{stack}}$ (pre-LPWCA) and $F_{\text{lp}}$ (post-LPWCA) under the same projections and normalization. Together, these metrics quantify cross-layer agreement, spatial stability, alignment with PWCA, and the smoothness of LWCA integration.

## C.2 HYPERPARAMETER SENSITIVITY ANALYSIS

To evaluate the robustness of CCRA to key hyperparameters, we conduct a sensitivity analysis on the attention projection dimension, $d_{\text{hidden}}$, and the Gaussian smoothing kernel size, $k$. These parameters respectively control the expressiveness of text-image alignment and the continuity of layer-wise semantic attention.

We vary $d_{\text{hidden}}$ across the set $\{64, 96, 128, 160, 192, 256, 320, 384\}$ and $k$ across $\{2, 3, 4, 5, 6, 7\}$, while evaluating accuracy on the GQA benchmark. As shown in Figure 7, the model demonstrates consistent performance across these settings. Notably, $d_{\text{hidden}} = 128$ and $k = 5$ yield the optimal trade-off between performance and stability, aligning with our default configuration.

## D QUALITATIVE EXAMPLES OF MODEL PREDICTIONS

To better illustrate the behavior of our CCRA model across diverse visual-language tasks, we present qualitative examples categorized into four groups: counting, OCR, object grounding, and binary (yes/no) questions. Each example includes an input image, a task-specific question, and the model's generated response.

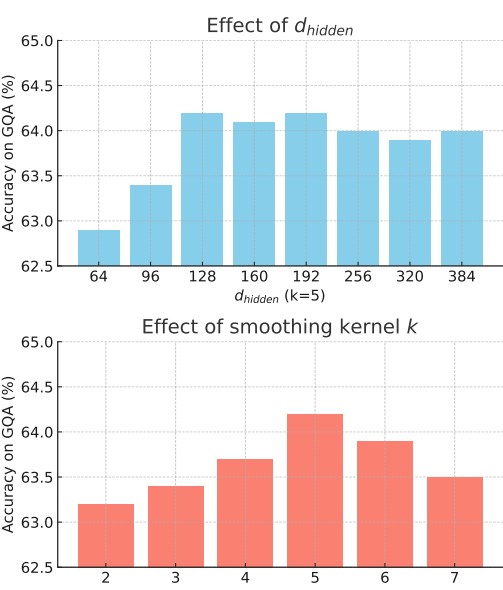

Figure 7: Sensitivity of CCRA on the GQA dataset with respect to hidden dimension $d_{\text{hidden}}$ and smoothing kernel size $k$.

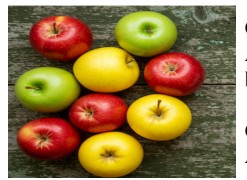

Q: How many green apples are there?
**A: There are 2 green apples in the image. You can spot them by their distinct bright green color. They stand out clearly among the red and yellow ones.**

Q: How many apples are there in total?
**A: In total, there are 9 apples in the image.**

(a) Counting Task

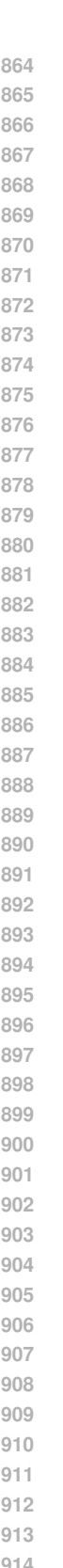

Q: Who edited the book?\nReference OCR token: Thackery, Pocket, Eccentric, Guide, &, to, Discredited, Diseases, COMPLETE, AA, by, MARK, ROBERTS, include:, JEFF, VANDERMEER, Edited, &, Contributing, KAGE, authors, BAKER, CORY, DOCTOROW, BISHOP, NEIL, MICHAEL, MIEVILLE, MOORCOCK, MOORE, others, CHINA, GAIMAN, ALAN, many, and\nAnswer the question using a single word or phrase.
**A: Jeff VanderMeer & Mark Roberts**

(b) OCR

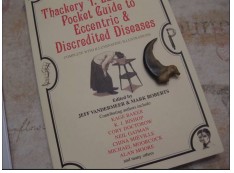

Q: Where is the backpack?

**A: The red backpack is located on the left side of the desk. It occupies the bottom-left corner of the image and is partially cropped, with one black strap extending outward**.

(c) Object grounding

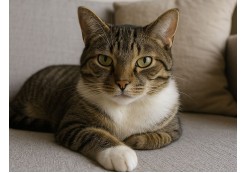

Q: Is the cat wearing a collar?

**A: No, the cat is not wearing any collar in this image.**

(d) Yes/No questions

Figure 8: Qualitative examples of CCRA model predictions across diverse tasks: (a) Counting (e.g., number of apples), (b) OCR (e.g., book editors), (c) Object Grounding (e.g., backpack location), and (d) Yes/No questions (e.g., presence of a collar).

As shown in Figure 8, the model demonstrates fine-grained understanding across various challenges, such as distinguishing apple colors and quantities, reading book titles and editor names, spatially grounding objects in cluttered scenes, and answering binary attribute-based questions. These results highlight the model's capability to effectively align vision and language under varying semantic demands.

