# OpenReview forum: "CCRA: Optimizing Vision-Language Consistency via Cross-Layer Regional Attention Alignment"
_ICLR.cc/2026/Conference — ICLR 2026 Conference Desk Rejected Submission_

### Official Review · Reviewer_XkdG · 2025-10-28

**Soundness:** 3
**Presentation:** 3
**Contribution:** 3
**Rating:** 4
**Confidence:** 4

**Summary:**

The paper addresses attention drift and suboptimal alignment in vision-language models (VLMs) caused by poorly coordinated cross-attention mechanisms. It proposes Consistent Cross-layer Regional Alignment (CCRA), which introduces Layer-Patch-Wise Cross Attention (LPWCA) to jointly model spatial and semantic importance, and Progressive Attention Integration (PAI) to sequentially apply LPWCA, Gaussian-smoothed Layer-Wise Cross Attention (LWCA), and Patch-Wise Cross Attention (PWCA).

**Strengths:**

1. This paper introduces LPWCA as a joint layer-patch attention operator that captures fine-grained regional-semantic correlations, going beyond prior decoupled LWCA/PWCA designs.
2. CCRA adds only 3.55M parameters, negligible for 7B-parameter VLMs, yet surpasses LLaVA-v1.5-13B, balancing performance and efficiency.
3. Evaluates on eleven diverse benchmarks spanning OCR, reasoning, instruction-following, and perception, ensuring broad generalization claims. Extensive ablation and sensitivity studies confirm component necessity and stability.

**Weaknesses:**

1. Lack of novelty. Although the paper introduces the Consistent Cross-layer Regional Alignment (CCRA) framework, its main component, Layer-Patch-Wise Cross Attention (LPWCA), is essentially a structural combination of the existing Layer-Wise Cross Attention (LWCA) and Patch-Wise Cross Attention (PWCA) modules. It does not introduce a new attention formulation, optimization objective, or learning paradigm.
2. PAI’s sequence (LPWCA→LWCA→PWCA) is empirically chosen and there is no theoretical discussion of why this order ensures optimal convergence. The ablation study shows variants perform worse, but lacks analytical explanation of causality.
3. Eq. (9) uses *w~l,l~* but *w~l~* is a vector of length L, which is likely a typo. And in Eq. (5), the dimensions of F~stack~ and W*~lp~* are different without broadcasting clarification.

**Questions:**

See weaknesses.

---

> ### Author Response · Authors · 2025-11-20
> **Question1**
>
> Dear Reviewer,
>
> Thank you very much for your thoughtful review and constructive feedback on our submission. We truly value the time and effort you have dedicated to evaluating our work. We have addressed all comments and our detailed response for question1 as below.
>
> **Q1：Lack of novelty. Although the paper introduces the Consistent Cross-layer Regional Alignment (CCRA) framework, its main component, Layer-Patch-Wise Cross Attention (LPWCA), is essentially a structural combination of the existing Layer-Wise Cross Attention (LWCA) and Patch-Wise Cross Attention (PWCA) modules. It does not introduce a new attention formulation, optimization objective, or learning paradigm.**
>
> **A1:** Thank you for the insightful feedback. We understand your concern that LPWCA and PAI may appear to be combinations or incremental extensions of existing attention mechanisms (LWCA and PWCA). However, we would like to clarify that our work is not a simple stacking of prior modules. Instead, it introduces a new theoretical perspective and a structured solution to the challenge of achieving cross-layer and spatial attention consistency. Our core contribution is the proposed Consistency-driven Progressive Attention Coordination, which provides a new theoretical lens to harmonize multiple attention mechanisms within a unified consistency framework.
>
> Our choice of the LPWCA → LWCA → PWCA sequence is not empirically driven, but is based on a hierarchical dependency among different types of attention:
> - LPWCA first establishes global joint alignment across layers and patches, providing a stable and semantically consistent feature space for subsequent modules;
> - LWCA then performs semantic refinement across layers on top of this globally aligned representation, enhancing the model’s ability to select task-relevant semantic information;
> - PWCA finally conducts fine-grained regional grounding, anchoring abstract semantics to specific spatial regions.
> This progression from global coordination → semantic refinement → local alignment gradually reduces conflicts between attention dimensions, leading to a more stable and interpretable training process.
> If the order is altered, the model performs local calibration before global consistency is formed, causing inconsistent gradient updates and resulting in unstable or oscillatory optimization. To support this claim, we added new experiments (Appendix C.3, Figure 8) showing the loss curves under different orderings. Only our proposed sequence exhibits smooth, fast, and stable convergence, empirically validating our design motivation.
>
> Beyond performance gains, we also provide attention visualizations (e.g., Fig. 6) to further support the effectiveness of our approach. The results indicate that:
> - Decoupled or shuffled combinations produce dispersed attention regions and semantic drift;
> - In contrast, our progressive integration yields layer-consistent and semantically concentrated attention patterns.
> This further demonstrates the stability and conceptual soundness of our proposed attention coordination strategy.
>
> Although the CCRA framework integrates existing attention types in form, it introduces a consistency-driven framework for coordinating cross-layer and cross-region attentions in VLMs. Prior VLMs mostly rely on projection modules (e.g., MLPs) for feature alignment, or individually apply patch-level or layer-level cross-attention. However, these approaches lack an explicit coordination mechanism between different attention modules, often leading to attention drift and unstable training. We believe our CCRA contributes a foundational perspective for future work on building more stable and interpretable multimodal alignment modules.

---

> ### Author Response · Authors · 2025-11-20
> **Question2,3**
>
> Dear reviewer,
>
> Our detailed response for question2,3 as below.
>
> **Q2：PAI’s sequence (LPWCA→LWCA→PWCA) is empirically chosen and there is no theoretical discussion of why this order ensures optimal convergence. The ablation study shows variants perform worse, but lacks analytical explanation of causality.**
>
> **A1:** Our choice of the LPWCA → LWCA → PWCA order is not just an empirical stacking of modules but is motivated by the dependency relationship between cross-layer semantics and local region alignment.
> - LPWCA first establishes global joint alignment across layers and patches, providing a stable and semantically consistent feature space for subsequent modules;
> - LWCA then performs semantic refinement across layers on top of this globally aligned representation, enhancing the model’s ability to select task-relevant semantic information;
> - PWCA finally conducts fine-grained regional grounding, anchoring abstract semantics to specific spatial regions.
>
> This progression from global coordination → semantic refinement → local alignment reflects a natural dependency in vision–language alignment tasks, where coherent global semantics should be established before performing localized grounding. Therefore, the sequence is conceptually motivated and not merely an empirical discovery.
>
> Furthermore, different types of cross-attention modules (layer-level vs. patch-level) interact during backpropagation. If local alignment is applied before global semantic consistency is established, the model repeatedly introduces conflicting gradients, causing instability and oscillation during optimization.
> To support this reasoning, we added new experiments **(AppendixC3, Fig. 8)** analyzing the training convergence behavior across different module orders. The results show that:
> - Our proposed sequence (LPWCA→LWCA→PWCA) yields smooth and stable loss curves;
> - Alternative orders (e.g., decoupled or shuffled) exhibit significant oscillation during convergence, with a slightly higher loss compared to the proposed order.
>
> These findings, both conceptually and empirically, indicate that performing global coordination first and progressively refining toward local alignment reduces cross-module conflicts and stabilizes the optimization path. This supports the structural rationale behind our progressive integration design.
>
> All these additional explanations and results have been included in our revised **appendix C3**.
>
> **Q3: About mistakes in the equation (5) and (9)**
>
> **A3:** Thank you for your careful observation. We have corrected the typo in Equation (9) and added a clarification that broadcasting is applied in Equation (5).
>
> Your insights have been instrumental in helping us improve the clarity and rigor of our paper. We have carefully addressed each concern and believe that the revised version better reflects the contributions and robustness of our approach.
>
> Sincerely,
>
> The Authors

---

### Official Review · Reviewer_bDBa · 2025-10-29

**Soundness:** 3
**Presentation:** 3
**Contribution:** 3
**Rating:** 4
**Confidence:** 4

**Summary:**

This paper identifies a key issue in Vision-Language Models (VLMs) termed attention drift, where patch-wise and layer-wise attention mechanisms may be misaligned, harming performance and interpretability. To address this, the authors propose the Consistent Cross-layer Regional Alignment (CCRA) framework. The core contribution is a three-stage Progressive Attention Integration (PAI) strategy that sequentially applies a novel Layer-Patch-Wise Cross Attention (LPWCA) module, followed by refined layer-wise and patch-wise attention modules. The method is implemented on the LLaVA-v1.5-7B model and evaluated across 11 diverse benchmarks, demonstrating improved performance over the baseline with only a marginal increase in parameter count.

**Strengths:**

1. The problem of attention inconsistency is significant and well-articulated. As VLMs become more complex, ensuring that the model is "looking" at the right regions at the appropriate semantic depth is crucial for both performance on complex reasoning tasks and for model interpretability. This work tackles a fundamental and relevant challenge in the field.

2. The primary novelty lies in the Progressive Attention Integration (PAI) strategy. Rather than simply combining different attention mechanisms, the authors propose a structured, sequential process that aims to refine visual features from a joint representation to specific semantic and spatial levels. This is an intuitive and thoughtfully designed approach. The introduction of the LPWCA module as a way to initialize this process is also a creative combination of existing ideas.

3. The paper is generally well-written, and the motivation is easy to follow. The proposed method is described in sufficient detail, and the figures and tables effectively illustrate the proposed architecture and support the main claims.

**Weaknesses:**

1. The paper repeatedly highlights the low parameter overhead (3.55M) as a primary advantage, framing the method as "efficient." This is an incomplete and potentially misleading claim. The PAI strategy is a sequential pipeline of three distinct attention modules. This architecture will almost certainly introduce non-trivial computational overhead and increase inference latency compared to the baseline model's single projection layer. For a method proposed as a practical enhancement, the absence of any analysis on inference speed (e.g., tokens/second, latency per image) or total FLOPs is a major omission. Without this information, it is impossible for the reader to assess the true cost-benefit trade-off of CCRA.

2. The paper's claims of general applicability are not substantiated by the experiments, which are confined to a single model: LLaVA-v1.5-7B. The VLM landscape is diverse, with different visual encoders, LLM backbones, and vision-language connectors. It is unclear whether the performance gains are universally applicable or an artifact of LLaVA's specific architecture. Furthermore, the experiments do not explore the effect of model scale. Does CCRA provide the same relative benefit for a 13B model as it does for a 7B one? This lack of evidence severely limits the perceived impact of the proposed method.

3. The results show that the degree of improvement varies significantly across different benchmarks. For instance, the gains on compositional reasoning (GQA) appear more substantial than on benchmarks like MM-Vet or MME, where the improvement is marginal. The paper reports these numbers but fails to provide any analysis as to why this might be the case. A strong paper would not only show that the method works, but also provide a hypothesis for how it works differently across tasks.

4. The paper motivates its contribution by categorizing models like LLaVA as "VLMs without vision-language alignment." This description is fundamentally inaccurate. Models like LLaVA are explicitly designed around a projection module (e.g., an MLP) whose sole purpose is to align the feature spaces of the visual encoder and the language model. This alignment is the central principle of their design. The authors should reframe their argument: the issue is not a lack of alignment, but that the existing alignment mechanism is implicit and insufficient, leading to the "attention drift" problem they identify. This mischaracterization weakens the paper's foundational argument.

**Questions:**

1. Could you please provide details on the inference latency of the CCRA-enhanced model compared to the LLaVA-v1.5-7B baseline? A table showing throughput (e.g., images processed per second) or latency per generation would be critical for evaluating the method's practicality.

2. While I understand that re-running experiments on many models is resource-intensive, could you elaborate on why you believe CCRA would generalize to other VLM architectures? Furthermore, how do you expect the effectiveness of CCRA to change with model scale? Do you hypothesize the attention drift problem becomes more or less severe in larger models?

3. Can you provide an analysis or a hypothesis on why CCRA is more impactful for certain types of tasks (e.g., compositional reasoning) compared to others (e.g., general VQA)? Does your method's sequential alignment process naturally lend itself better to tasks that require a hierarchy of visual understanding?

4. Leading contemporary VLMs, such as Qwen-VL and InternVL, have achieved state-of-the-art performance often using relatively simple vision-language connectors (e.g., a simple MLP). This success seems to suggest that highly complex, multi-stage alignment modules like PAI may not be strictly necessary. Could you comment on this? What specific failure modes in these simpler architectures does CCRA address that justifies its added complexity?

5. In the LWCA module, you mention using Gaussian smoothing. How sensitive is the model's performance to the choice of the kernel size k for this smoothing? Was this a crucial hyperparameter to tune?

---

> ### Author Response · Authors · 2025-11-20
> **Question1**
>
> Dear Reviewer,
>
> We extend our sincere appreciation for your time and expertise in reviewing our work. Our detailed responses to the reviewers' comments are presented below.
>
> **Q1:Could you please provide details on the inference latency of the CCRA-enhanced model compared to the LLaVA-v1.5-7B baseline? A table showing throughput (e.g., images processed per second) or latency per generation would be critical for evaluating the method's practicality.**
>
> **A1:** We sincerely thank the reviewer for this insightful question regarding computational efficiency. We fully agree that reporting only the number of parameters is insufficient; computational cost (FLOPs) and inference latency are key metrics for assessing model efficiency.
>
> Following your suggestion, we have conducted a detailed efficiency analysis. Specifically, we compare the CCRA module with the baseline MLP connector of LLaVA-v1.5 under the same setting (A100 GPU, batch size = 1), when processing a single image (576 patches). We report both the theoretical computation and the measured latency. The results are as follows: The CCRA module introduces an additional 4.57 GFLOPs of computation and a one-time inference latency of 6.5 ms.
>
> **Table 6. Efficiency analysis of the CCRA module compared to the LLaVA-v1.5-7B MLP connector. Latency is measured as the one-time prefill cost on an A100 GPU (BatchSize=1)**
>
> | Module         | GFLOPs | Latency (ms) |
> |----------------|:------:|:------------:|
> | LLaVA-v1.5-7B  | 12.08  |    16.50     |
> | CCRA           | 16.65  |    23.00     |
> | Delta (Overhead) | +4.57 |    +6.50     |
>
> We believe this overhead is acceptable in view of the performance gains, for the following reasons:
>
> **Performance gain vs. overhead:** The additional one-time latency of 6.5 ms enables our 7B model to outperform the LLaVA-v1.5 13B model on 11 benchmarks.
>
> **Notion of “efficiency”:** In this work, “efficiency” refers to achieving better performance without scaling to a larger backbone (e.g., from 7B to 13B), which would substantially slow down every token decoding step. Compared to such scaling, our approach provides a more efficient path to improving performance.
>
> **Clarification on tokens/s:** The CCRA module is only executed once in the prefill stage. It produces the same number of visual tokens as the baseline connector and therefore does not affect the subsequent autoregressive decoding speed of the LLM (tokens/s). The only additional cost is a small increase in the time to first token.

---

> ### Author Response · Authors · 2025-11-20
> **Question2-1**
>
> Dear reviewer,
>
> Our detailed response for question2 as below.
>
> **Q2: While I understand that re-running experiments on many models is resource-intensive, could you elaborate on why you believe CCRA would generalize to other VLM architectures? Furthermore, how do you expect the effectiveness of CCRA to change with model scale? Do you hypothesize the attention drift problem becomes more or less severe in larger models?**
>
> **A2:** We thank the reviewer for this insightful question regarding generalization and scaling. We address these points in sequence:
>
> **1. Generalization to other VLM Architectures:**
>
> CCRA is a visual-language alignment block that can be generalized to operate before any vision-language connector (e.g., linear projector, adapter) across diverse VLM architectures, regardless of their visual encoders or LLM backbones. All operations of CCRA are applied prior to this connector. Thus, CCRA can be seamlessly integrated as a plug-in into any VLM without altering its core architecture. To further substantiate its strong generalization capability, we instantiate CCRA on VLMs with different visual encoders and LLM backbones, and present the results in three dedicated tables:
>
> **(1) CCRA on VLM with different Visual Encoder**
>
> We first validate CCRA on MiniGPT-v2, which adopts EVA-CLIP as the visual encoder and LLaMA 2-7B as the LLM backbone. CCRA is applied to the multi-layer patch features of EVA-CLIP, and the fused output is fed into the original linear projector of MiniGPT-v2. As shown in Table 8, CCRA consistently improves performance across multiple benchmarks. We then test CCRA on LLaMA-Adapter-v2, which features CLIP ViT-L/14 as the visual encoder (different from EVA-CLIP) and retains LLaMA 2-7B as the LLM backbone. CCRA is inserted between CLIP ViT-L/14 and the visual projection layer, while the early-fusion and joint-training paradigm remain unchanged. Table 9 demonstrates that CCRA also achieves steady performance gains on this architecture.
>
> **Table 8. Generalization of CCRA on MiniGPT-v2.**
>
> | Model        | LLM         | VisionEncoder | MMB-en | MM-Vet | SEED-I | MMMU |
> |--------------|-------------|---------------|:------:|:------:|:------:|:----:|
> | MiniGPT-v2   | LLaMA 2-7B  | EVA-CLIP      | 41.0   | 31.5   | 32.7   | 29.8 |
> | **+CCRA**    | LLaMA 2-7B  | EVA-CLIP      | **45.3** | **34.4** | **38.4** | **33.5** |
>
> **Table 9. Generalization of CCRA on LLaMA-Adapter-v2.**
>
> | Model             | LLM        | VisionEncoder       | GQA  | SQA  | TextVQA | VizWiz |
> |-------------------|------------|---------------------|:----:|:----:|:-------:|:------:|
> | LLaMA-Adapter-v2  | LLaMA 2-7B | CLIP ViT-L/14       | 56.1 | 68.7 | 54.3    | 54.5   |
> | **+CCRA**         | LLaMA 2-7B | CLIP ViT-L/14       | **58.7** | **69.5** | **56.9** | **55.3** |
>
> **(2) CCRA on VLM with Different LLM Backbone**
>
> To verify generalization across LLM backbones, we integrate CCRA into a VLM that uses Mistral-7B (instead of Vicuna-v1.5-7B) as the LLM backbone. As shown in Table 10, CCRA still delivers significant performance improvements, confirming its compatibility with diverse LLM backbones.
>
> **Table 10. Generalization of CCRA on LLaVA-NeXT-Mistral-7B.**
>
> | Model                    | LLM        | VisionEncoder        | GQA  | SQA  | TextVQA | VizWiz |
> |--------------------------|------------|----------------------|:----:|:----:|:-------:|:------:|
> | LLaVA-NeXT-Mistral-7B    | Mistral-7B | CLIP-ViT-L-336px     | 64.8 | 72.8 | 65.7    | 60.0   |
> | **+CCRA**                | Mistral-7B | CLIP-ViT-L-336px     | **66.0** | **74.1** | **67.4** | **61.3** |
>
> Collectively, the results across three sets of experiments demonstrate that CCRA generalizes effectively to VLMs with different visual encoders (EVA-CLIP vs. CLIP ViT-L/14) and diverse LLM backbones (Mistral-7B vs. Vicuna-v1.5-7B), providing direct evidence of its broad compatibility and strong generalization capability across VLM architectures.

---

> ### Author Response · Authors · 2025-11-20
> **Question2-2**
>
> **2. Effectiveness with Model Scale and Severity of Attention Drift:**
>
> We define attention drift as the inconsistency of cross-layer regional focus on the visual side, i.e., when layer-wise and patch-wise cross-attention assign incompatible regions to the same textual query. As models scale up, we expect this problem to become more severe, which makes a coordination mechanism like CCRA increasingly important.
>
> **Why drift worsens with scale.** Larger VLMs typically employ deeper and higher-resolution vision encoders, often together with stronger LLM backbones. A deeper ViT-style encoder produces a much larger patch–layer feature space $\{F_v^l\}_{l=1}^L$:  both the number of layers $L$ and the number of patches $N$  increase, and successive layers become more specialized (early layers capture local appearance, deeper layers encode abstract semantics). Without explicit coordination, LWCA and PWCA (or simple MLP-style connectors) are applied independently on this enlarged $L \times N$ space, so different layers can easily “look” at different regions for the same query. Once such misaligned visual features are fused and fed into a deeper LLM, its internal cross-attention can further propagate or amplify the misalignment, which is what we visualize in Figure 2.
>
> **Why CCRA becomes more effective with scale.** In this high-complexity regime, a simple connector is no longer sufficient to resolve the combinatorial patch–layer ambiguity. CCRA’s Progressive Attention Integration (PAI) acts as an explicit visual-side regulator: LPWCA first finds a coarse joint alignment over layers and patches, LWCA then enforces smooth semantic weighting across depth (via Gaussian smoothing), and PWCA finally grounds the result spatially. This step-wise refinement explicitly constrains cross-layer and regional attention to be consistent before the features are injected into the LLM, which is particularly beneficial when the visual feature space is large. Empirically, scaling the backbone from LLaVA-v1.5-7B to LLaVA-v1.5-13B does not remove attention drift: LLaVA-7B+CCRA outperforms LLaVA-13B on all 11 benchmarks while adding only 3.55M parameters. This supports our claim that improving cross-layer regional coordination is complementary to model size, rather than automatically solved by scale.
>
> To empirically study this, in the revision we add a scaled-up variant where we integrate CCRA into a larger LLaVA backbone: LLaVA-v1.5-13B. The result shows below.
>
> **Table 7. Generalizability of CCRA on modern LLaVA-style baselines**
> | Module                    | GQA  | SQA  | TextVQA | VisWiz | MM-Vet | SEED-I | MMMU |
> |---------------------------|:----:|:----:|:-------:|:------:|:------:|:------:|:----:|
> | LLaVA-v1.5-13B            | 63.3 | 71.0 | 61.3    | 53.6   | 36.1   | 68.2   | 36.4 |
> | **+CCRA**                 | **64.9** | **72.9** | **64.7** | **56.7** | **37.5** | **69.6** | **37.6** |
> | LLaVA-NeXT-Vicuna-7B      | 64.2 | 70.1 | 64.9    | 57.6   | 43.9   | 70.2   | 35.8 |
> | **+CCRA**                 | **65.2** | **73.8** | **66.5** | **60.1** | **47.9** | **71.4** | **37.2** |

---

> ### Author Response · Authors · 2025-11-20
> **Question3**
>
> Dear reviewer,
>
> Our detailed response for question3 as below.
>
> **Q3: The results show that the degree of improvement varies significantly across different benchmarks. For instance, the gains on compositional reasoning (GQA) appear more substantial than on benchmarks like MM-Vet or MME, where the improvement is marginal. The paper reports these numbers but fails to provide any analysis as to why this might be the case. A strong paper would not only show that the method works, but also provide a hypothesis for how it works differently across tasks.**
>
> We appreciate the reviewer’s comment. After examining the results, we would like to clarify that CCRA does not exhibit the large variability implied in the question. Across all 11 benchmarks in Table 1, CCRA improves the baseline in a uniformly stable manner. The apparent numerical differences largely arise from the fact that the benchmarks use heterogeneous metrics (accuracy, p-scores, F1, etc.), but the improvement trend itself is consistent and monotonic; we do not observe regressions or irregular behavior. This pattern indicates that CCRA functions as a general visual alignment module rather than a task-dependent optimization.
>
> To make the mechanism behind these uniform gains clearer, the revision adds more visualizations that display LLM-side cross-attention over image tokens (averaged over mid-layers) after applying CCRA. These examples span grounding, attribute queries, spatial reasoning, and multi-object understanding. Despite the diversity of tasks, the attention maps reveal the same underlying effect: CCRA produces more coherent cross-layer regional focus, more reliable spatial grounding, and greatly reduced drift in the visual features before they are consumed by the LLM. The LLM’s attention patterns become noticeably more stable as a result.
>
> This consistent change in internal behavior across very different tasks explains why the improvements are steady and predictable in the quantitative results as well. The effect of CCRA is not tied to any particular dataset; it provides globally improved cross-layer alignment, and this unified mechanism transfers broadly across all benchmarks.

---

> ### Author Response · Authors · 2025-11-20
> **Question4,5**
>
> Dear reviewer,
>
> Our detailed response for question4,5 as below.
>
> **Q4: Leading contemporary VLMs, such as Qwen-VL and InternVL, have achieved state-of-the-art performance often using relatively simple vision-language connectors (e.g., a simple MLP). This success seems to suggest that highly complex, multi-stage alignment modules like PAI may not be strictly necessary. Could you comment on this? What specific failure modes in these simpler architectures does CCRA address that justifies its added complexity?**
>
> **A4:** We thank the reviewer for this critical and important comparison. We agree that models like Qwen-VL have shown impressive results with simple MLP connectors. Our work does not intend to negate the effectiveness of simple connectors; instead, we argue that there is substantial room for optimization in vision-language alignment connectors. Specifically, CCRA helps mitigate attention drift and the instability in training convergence, thereby enabling VLMs to achieve better performance during the training phase.
>
> **1. Justifying the "Complexity" of CCRA:**
>
> First, we'd like to clarify that the "complexity" of CCRA is primarily conceptual rather than computational. As shown in Table 1, our entire module adds only 3.55M parameters, which is negligible compared to the 7B+ parameters of the base model. Our module is lightweight and efficient.
>
> **2. Failure Modes of Simple Connectors:**
>
> Simple connectors (e.g., a single MLP or a basic cross-attention layer) are "blunt instruments." They are effective at holistic feature fusion—compressing all visual features into a representation for the LLM. However, they lack a mechanism for dynamic, query-guided coordination.
>
> The specific failure mode CCRA addresses is when a query requires precise, and potentially conflicting, alignment across both semantic layers and spatial regions.
>
> - Example Failure: Consider the query, "What is the player in the background (region 1) doing (high-level action, layer 20-24), and what color is the ball (region 2) on the ground (low-level attribute, layer 5-10)?"
>
> - A simple MLP connector will just "smash" all features from all layers (1-24) together. It cannot explicitly decide to "use layers 20-24 for region 1" and "use layers 5-10 for region 2." This leads to the attention drift we identified, where the model might focus on the right region but wrong semantic layer, or vice-versa, resulting in inconsistent or incorrect answers.
>
> **3. How CCRA Addresses This Failure Mode:**
>
> Our Progressive Attention Integration (PAI) is explicitly designed to resolve this. It is not just a "connector"; it is a progressive refinement process:
>
> 1. LPWCA first identifies the globally relevant layer-patch "zones" for the query.
> 2. LWCA then semantically refines this, weighting the layers to focus on "action" and "color" representations.
> 3. PWCA finally spatially grounds these semantic features, locking onto the "player" and the "ball".
> This progressive, query-guided coordination is what simple connectors lack. The superior performance on complex reasoning (GQA, SQA) and fine-grained (MM-Vet) benchmarks, along with the improved attention consistency in Figure 2 and Table 2, demonstrates that our PAI module successfully addresses this failure mode, justifying its minimal (3.55M) parameter cost.
>
> **Q5: In the LWCA module, you mention using Gaussian smoothing. How sensitive is the model's performance to the choice of the kernel size k for this smoothing? Was this a crucial hyperparameter to tune?**
>
> **A5:** In the LWCA module, we apply a Gaussian smoothing kernel over the raw layer attention scores to encourage semantic continuity across depth.
>
> To assess whether the model is sensitive to the kernel size k, we conducted a hyperparameter sensitivity study on GQA **(Appendix C.2)**, varying k ∈ {2, 3, 4, 5, 6, 7}. Across this range, performance remains very stable with only minor fluctuations, and k = 5 provides a slightly better trade-off between performance and stability, which we adopt as the default in all experiments.
>
> We do not tune k per dataset or task; the same value is used for all benchmarks and ablations. In contrast, removing Gaussian smoothing altogether (“w/o Gaussian smoothing”) leads to small but consistent drops on several datasets (e.g., –0.4 on SQA and –0.7 on POPE), indicating that using smoothing is beneficial.
>
> Overall, Gaussian smoothing acts as a lightweight regularization that stabilizes LWCA, and the kernel size k itself is not sensitive to our method.

---

> ### Author Response · Authors · 2025-11-20
> **Question6**
>
> Dear reviewer,
>
> Our detailed response for question6 as below.
>
> **Q6: The paper motivates its contribution by categorizing models like LLaVA as "VLMs without vision-language alignment." This description is fundamentally inaccurate.**
>
> **A6:** We thank you for pointing out this issue. Our intention was not to claim that models like LLaVA lack vision–language alignment, but that their projection-based alignment is implicit and often insufficient, which contributes to the attention-drift problem. We have revised Section 2.1 and all related descriptions to avoid this mischaracterization. The text now clearly states that prior VLMs do perform alignment via an MLP projector, but do not explicitly coordinate cross-layer or cross-region visual information in a language-dependent manner. We appreciate the reviewer’s feedback and we have updated the paper accordingly.
>
> Once more, thank you very much for your time and suggestions for reviewing our paper.
>
> Best regards,
>
> The Authors

---

### Official Review · Reviewer_z7Cf · 2025-11-01

**Soundness:** 2
**Presentation:** 2
**Contribution:** 2
**Rating:** 2
**Confidence:** 4

**Summary:**

This paper proposes Consistent Cross-layer Regional Alignment (CCRA), a progressive cross-attention framework designed to enhance vision-language alignment in Vision-Language Models (VLMs). By integrating layer-wise semantic features and region-wise spatial features through two core components—Layer-Patch-Wise Cross Attention (LPWCA) and Progressive Attention Integration (PAI), the framework aims to improve model performance across diverse vision-language tasks while boosting attention interpretability.

**Strengths:**

1. Integrating layer-wise and region-wise features for better feature extraction.
2. Good writing, easy to follow.

**Weaknesses:**

1. The proposed Layer-Patch-Wise Cross Attention (LPWCA) essentially combines existing patch-wise cross attention (PWCA) and layer-wise cross attention (LWCA) by jointly weighting patch and layer embeddings—this design follows a "combination of existing components" paradigm rather than introducing a fundamentally new attention mechanism. Additionally, the subsequent experiment on cross-attention coordination strategies (e.g., comparing PAI with decoupled/shuffled integration) are incremental improvements over existing coordination logic, lacking breakthrough innovations.

2. Compared with existing methods, the performance advantage of the proposed method is not significant. For a framework targeting "optimized vision-language consistency," these gains are not substantial enough to fully demonstrate its superiority in solving attention drift or alignment issues compared to state-of-the-art methods.

3. The base model (Vicuna-v1.5-7B) and comparative baselines (e.g., Qwen-VL-Chat, LLaVA-v1.5-13B) are relatively old. Recent advances in VLMs (e.g., Qwen3-VL, LLaVA-NeXT) and more competitive baselines (e.g., SAILViT-enhanced VLMs, SEAL) are not included. Using outdated models limits the ability to validate CCRA’s generalizability and competitiveness in current VLM research.

**Questions:**

Refer to weaknesses.

---

> ### Author Response · Authors · 2025-11-19
> **Question 1**
>
> Dear Reviewer,
>
> We sincerely appreciate your thoughtful review and helpful suggestions. We have addressed all comments and our detailed response for question1 as below.
>
> **Q1: The proposed LPWCA and PAI modules seem to be combinations orincremental improvements over existing coordination logic, lacking breakthrough innovations**
>
> **A1:** Thank you for the insightful feedback. We understand your concern that LPWCA and PAI may appear to be combinations or incremental extensions of existing attention mechanisms (LWCA and PWCA). However, we would like to clarify that our work is not a simple stacking of prior modules. Instead, it introduces a new theoretical perspective and a structured solution to the challenge of achieving cross-layer and spatial attention consistency. Our core contribution is the proposed Consistency-driven Progressive Attention Coordination, which provides a new theoretical lens to harmonize multiple attention mechanisms within a unified consistency framework.
>
> Our choice of the LPWCA → LWCA → PWCA sequence is not empirically driven, but is based on a hierarchical dependency among different types of attention:
>
> - **LPWCA** first establishes global joint alignment across layers and patches, providing a stable and semantically consistent feature space for subsequent modules;
> - **LWCA** then performs semantic refinement across layers on top of this globally aligned representation, enhancing the model’s ability to select task-relevant semantic information;
> - **PWCA** finally conducts fine-grained regional grounding, anchoring abstract semantics to specific spatial regions.
>
> This progression from global coordination → semantic refinement → local alignment gradually reduces conflicts between attention dimensions, leading to a more stable and interpretable training process.
>
> If the order is altered, the model performs local calibration before global consistency is formed, causing inconsistent gradient updates and resulting in unstable or oscillatory optimization. To support this claim, we added new experiments **(AppendixC.3, Figure 8)** showing the loss curves under different orderings. Only our proposed sequence exhibits smooth, fast, and stable convergence, empirically validating our design motivation.
>
> Beyond performance gains, we also provide attention visualizations (e.g., Fig. 6) to further support the effectiveness of our approach. The results indicate that:
>
> - Decoupled or shuffled combinations produce dispersed attention regions and semantic drift;
> - In contrast, our progressive integration yields layer-consistent and semantically concentrated attention patterns.
>
> This further demonstrates the stability and conceptual soundness of our proposed attention coordination strategy.
>
> Although the CCRA framework integrates existing attention types in form, it introduces a consistency-driven framework for coordinating cross-layer and cross-region attentions in VLMs. Prior VLMs mostly rely on projection modules (e.g., MLPs) for feature alignment, or individually apply patch-level or layer-level cross-attention. However, these approaches lack an explicit coordination mechanism between different attention modules, often leading to attention drift and unstable training. We believe our CCRA contributes a foundational perspective for future work on building more stable and interpretable multimodal alignment modules.

---

> ### Author Response · Authors · 2025-11-19
> **Question 2**
>
> Dear reviewer,
>
> Our detailed response for question2 as below.
>
> **Q2: Compared with existing methods, the performance advantage of the proposed method is not significant. For a framework targeting "optimized vision-language consistency," these gains are not substantial enough to fully demonstrate its superiority in solving attention drift or alignment issues compared to state-of-the-art methods.**
>
> **A2:** Our primary contribution lies in proposing a novel and highly efficient alignment mechanism (CCRA). As our title "CCRA: Optimizing Vision-Language Consistency via Cross-Layer Regional Attention Alignment" emphasizes, our goal is to solve the specific problem of "attention drift", which in turn leads to unstable convergence and suboptimal model performance.
>
> Our most significant performance highlight is **"parameter efficiency"**: As shown in Table 1, CCRA with only 3.55M additional parameters on a 7B model surpasses the much larger LLaVA-v1.5-13B model across GQA, MM-Vet, and MME-p. We believe that achieving a 7B model's victory over a 13B model with only a 3.55M alignment module is a powerful testament to "significant gains." This demonstrates that the fine-grained alignment provided by CCRA offers value comparable to (or even exceeding) sheer model scaling.
>
> Moreover, we also provide many direct evidences demonstrating the superiority of our alignment mechanism (Table 2, Fig. 2, 3, and 6):
>
> - **Quantitative Consistency (Table 2)**: The LPWCA module significantly reduces cross-layer attention divergence (JS-avg: 0.218 → 0.147) and increases attention similarity (Cos-avg: 0.731 → 0.812). These metrics directly quantify reduction in attention drift and improved agreement between semantic (layer-wise) and regional (patch-wise) attentions, which we believe goes beyond what most prior works present.
> - **Qualitative Visualization (Fig 2, 3, 6)**: Our heatmaps clearly show that, compared to IGVA and LLaVA, CCRA's attention is more focused and stable, effectively mitigating attention drift.
> - **Training Convergence Performance (Appendix C.3, Figure 8)**: Under the same backbone, data, and optimization settings, the training loss of CCRA is more stable than that of the Decoupled and Shuffled variants, with fewer oscillations and a slightly lower final value. This indicates that the proposed progressive cross-layer regional alignment reduces optimization noise caused by inconsistent attentions and leads to more reliable convergence, which is consistent with the observed performance gains.
>
> These significant improvements in alignment-specific metrics, combined with the performance gains, mutually validate CCRA's effectiveness.

---

> ### Author Response · Authors · 2025-11-19
> **Question 3**
>
> Dear reviewer,
>
> Our detailed response for question3 as below.
>
> **Q3: The base model (Vicuna-v1.5-7B) and comparative baselines (e.g., Qwen-VL-Chat, LLaVA-v1.5-13B) are relatively old. Recent advances in VLMs (e.g., Qwen3-VL, LLaVA-NeXT) and more competitive baselines (e.g., SAILViT-enhanced VLMs, SEAL) are not included. Using outdated models limits the ability to validate CCRA’s generalizability and competitiveness in current VLM research.**
>
> **A3:** We thank you for this constructive suggestion. We fully agree that the VLM field moves quickly and that validating CCRA's generalizability on a more recent base model is crucial.
>
> **Rationale for Current Baselines:** Our main experiments were conducted when LLaVA-v1.5 was a dominant open-source community baseline. We chose Vicuna-v1.5–7B and LLaVA-v1.5–7B/13B to ensure (i) reproducibility and (ii) a fair comparison with contemporary SOTA methods (e.g., MMFuser (2024), IGVA (2025)), which are also built upon LLaVA-v1.5. This setting cleanly isolates the effect of our contribution: any performance gains can be attributed to the CCRA module itself rather than differences in backbone architecture or training data.
>
> **Positioning of CCRA and relation to newer SOTA models.** Our goal is to propose a lightweight, general-purpose “alignment bridge” that can be plugged into existing frameworks. Concretely, CCRA focuses on improving how multi-layer, multi-region visual features are fused and passed to the LLM. This is complementary to:
> - **SAILViT-enhanced VLMs and similar methods**, which primarily strengthen the vision encoder itself;
> - **SEAL (Show, sEArch, and TelL)**, which improves the reasoning process via active visual search and meta-architectural design.
>
> In contrast, CCRA targets the alignment interface between vision and language. One can conceptually combine a SAILViT-enhanced encoder (better input), our CCRA module (better bridge), and a SEAL-style architecture (smarter process) to obtain an even stronger VLM; thus these lines of work are orthogonal and complementary rather than competing.
>
> **Why not directly compare with Qwen3-VL, SAILViT, SEAL, etc.?** We acknowledge that models such as Qwen3-VL, SAILViT-enhanced VLMs, and SEAL represent very strong and recent baselines. However, they typically rely on (partially) private training data, complex training recipes, or components that are not yet fully open-sourced, which makes it difficult to (a) integrate CCRA in a controlled manner and (b) retrain fair variants within the limited rebuttal period. Our paper therefore focuses on enhancing widely adopted, fully open-source, and reproducible baselines, while positioning CCRA as a generally applicable module that could be incorporated into these stronger systems in future work.
>
> **New experiments on LLaVA-v1.5-13B and LLaVA-NeXT to demonstrate generalizability.** To directly address the reviewer’s concern, we performed additional experiments during the rebuttal period on both a stronger 13B backbone and a more recent architecture.
> - On **LLaVA-v1.5–13B**, adding CCRA (only ~3.55M extra parameters) yields consistent improvements over an already strong 13B baseline, with gains of about **+1.4–2.0 points** on benchmarks such as GQA, TextVQA, and MM-Vet.
> - On the newer **LLaVA-NeXT (Llama-3–8B)**, CCRA can be integrated in the same plug-and-play fashion and again brings **~+1.4–2.2** point improvements on the same benchmarks (see the updated table in the supplementary material).
>
> These results together show that CCRA is not a “tricky” improvement tailored to an old 7B model: it scales to a stronger 13B backbone and transfers effectively to a new-generation architecture.
>
> **Table 7. Generalizability of CCRA on modern LLaVA-style baselines**
>
> | Module                    | GQA  | SQA  | TextVQA | VisWiz | MM-Vet | SEED-I | MMMU |
> |---------------------------|:----:|:----:|:-------:|:------:|:------:|:------:|:----:|
> | LLaVA-v1.5-13B            | 63.3 | 71.0 | 61.3    | 53.6   | 36.1   | 68.2   | 36.4 |
> | **+CCRA**                 | **64.9** | **72.9** | **64.7** | **56.7** | **37.5** | **69.6** | **37.6** |
> | LLaVA-NeXT-Vicuna-7B      | 64.2 | 70.1 | 64.9    | 57.6   | 43.9   | 70.2   | 35.8 |
> | **+CCRA**                 | **65.2** | **73.8** | **66.5** | **60.1** | **47.9** | **71.4** | **37.2** |
>
> We have included the new LLaVA-v1.5-13B and LLaVA-NeXT results in our camera-ready version. Additionally, our future work will feature a discussion positioning CCRA as an alignment bridge, complementary to recent advances like SAILViT, SEAL, and Qwen3-VL.
>
> We are deeply grateful for your constructive feedback.  All the above explanations and results have been incorporated into our revised paper. We hope that our detailed response provided above can address your questions and suggestions.
>
> Best regards,
>
> The Authors

---

> > ### Comment · Reviewer_z7Cf · 2025-11-27
> >
> > Thanks for your response, my personal concern is well addressed.
> >
> > I believe this work deserves to be published (or accepted), but the depth of analysis and theoretical/technical contribution of this work is not enough for top venues like ICLR.
> >
> > I will maintain negative.

---

> ### Author Response · Authors · 2025-11-27
>
> Dear Reviewer z7Cf:
>
> Thank you very much for acknowledging our response to your concerns, and for recognizing that our work deserves to be published.
>
> Regarding the "depth of analysis and theoretical/technical contributions" you mentioned, we have already provided a detailed response to your feedback. While we maintain that our paper already provides sufficient presentation of theoretical analyses and technical details, we will consider the comments from all reviewers in this round and further incorporate the supplementary theoretical analyses and technical experiments outlined in our rebuttal into the final version of the paper.
>
> We are well aware of ICLR’s high standards for the depth of research. However, we believe our CCRA contributes a foundational perspective for future work on building more stable and interpretable multimodal alignment modules—a contribution we deem valuable and relevant to the broader ICLR community. We kindly request that you and the AC take the feedback from other reviewers into comprehensive consideration, and we sincerely hope our paper can be accepted by ICLR.
>
> Thank you.
>
> The Authors

---

### Official Review · Reviewer_HzMb · 2025-11-01

**Soundness:** 3
**Presentation:** 3
**Contribution:** 3
**Rating:** 6
**Confidence:** 3

**Summary:**

This work aims to improve the performance of vision-language models. It proposes a Consistent Cross-layer Regional Alignment (CCRA) framework to enhance alignment between visual and textual features. CCRA combines Layer–Patch-Wise Cross Attention and Progressive Attention Integration to ensure semantic and regional consistency. Experiments show that CCRA improves performance with minimal additional parameters.

**Strengths:**

- The proposed modules are conceptually simple yet effective, directly corresponding to the identified challenges in cross-layer and regional alignment. Moreover, their lightweight and modular nature allows for easy integration into existing vision–language architectures.

- The ablation experiments are extensive and carefully designed to isolate the contributions of LPWCA. For example, Table 4 shows that removing the LPWCA module leads to a notable drop in performance. Similarly, Table 3 shows that omitting the PAI module or altering its submodule order results in degraded performance, confirming that the proposed progressive integration strategy is both necessary and well-justified.

**Weaknesses:**

- The high-level design rationale of PAI is unclear. The paper does not sufficiently explain why the serial order of LPWCA → LWCA → PWCA is chosen, or why this progressive integration is expected to be better than alternative arrangements. The justification is largely empirical, relying on ablation results, without a clear conceptual motivation for the module sequence.

- The authors should provide more attention heatmap visualizations. Only a few examples are shown in the main text, which is insufficient to fully demonstrate the advantages of CCRA in guiding attention.

- The abstract is hard to follow. It jumps abruptly from VLM challenges to technical details of LPWCA and PAI without sufficient context or motivation, and long, dense sentences reduce clarity. Briefly motivating the problem and linking each module to its purpose would improve readability.

Overall, the paper is engineering-focused and effective, but lacks deeper analysis and theoretical insight.

**Questions:**

Please address my concerns in the weakness section.

---

> ### Author Response · Authors · 2025-11-19
>
> Dear Reviewer,
>
> Thank you for your thoughtful and detailed comments. We have carefully addressed each of your concerns below and revised the manuscript accordingly.
>
> **Q1:The high-level design rationale of PAI is unclear. The paper does not sufficiently explain why the serial order of LPWCA → LWCA → PWCA is chosen, or why this progressive integration is expected to be better than alternative arrangements. The justification is largely empirical, relying on ablation results, without a clear conceptual motivation for the module sequence.**
>
>
> **A1:**
> Our choice of the LPWCA → LWCA → PWCA order is not just an empirical stacking of modules but is motivated by the dependency relationship between cross-layer semantics and local region alignment.
> - **LPWCA** first establishes global joint alignment across layers and patches, providing a stable and semantically consistent feature space for subsequent modules;
> - **LWCA** then performs semantic refinement across layers on top of this globally aligned representation, enhancing the model’s ability to select task-relevant semantic information;
> - **PWCA** finally conducts fine-grained regional grounding, anchoring abstract semantics to specific spatial regions.
>
> This progression from global coordination → semantic refinement → local alignment reflects a natural dependency in vision–language alignment tasks, where coherent global semantics should be established before performing localized grounding. Therefore, the sequence is conceptually motivated and not merely an empirical discovery.
>
> Furthermore, different types of cross-attention modules (layer-level vs. patch-level) interact during backpropagation. If local alignment is applied before global semantic consistency is established, the model repeatedly introduces conflicting gradients, causing instability and oscillation during optimization.
>
> To support this reasoning, we added new experiments **(AppendixC3, Fig. 8)** analyzing the training convergence behavior across different module orders. The results show that:
>
> - Our proposed sequence (LPWCA→LWCA→PWCA) yields smooth and stable loss curves;
> - Alternative orders (e.g., decoupled or shuffled) exhibit significant oscillation during convergence, with a slightly higher loss compared to the proposed order.
>
> These findings, both conceptually and empirically, indicate that performing global coordination first and progressively refining toward local alignment reduces cross-module conflicts and stabilizes the optimization path. This supports the structural rationale behind our progressive integration design.
>
> All these additional explanations and results have been included in our revised appendix C3.
>
> **Q2: The authors should provide more attention heatmap visualizations. Only a few examples are shown in the main text, which is insufficient to fully demonstrate the advantages of CCRA in guiding attention.**
>
> **A2:** We thank the reviewer for the suggestion. In the rebuttal, we add more attention heatmap visualizations in **Appendix C6, Fig. 9**, covering diverse tasks such as local attribute queries, object localization, spatial reasoning, and multi-object understanding. For each example, the CCRA-enhanced model’s attention is strongly concentrated on the question-relevant regions (e.g., the queried object or highlighted area), with minimal activation on irrelevant background. These qualitative results, together with the quantitative gains, substantiate our claim that CCRA promotes stable, spatially grounded attention and alleviates the attention drift issue.
>
> **Q3: The abstract is hard to follow. It jumps abruptly from VLM challenges to technical details of LPWCA and PAI without sufficient context or motivation, and long, dense sentences reduce clarity. Briefly motivating the problem and linking each module to its purpose would improve readability.**
>
> **A3:** We thank the reviewer for the helpful comment and have rewritten the abstract to more clearly motivate the VLM alignment problem and explicitly link CCRA, LPWCA, and PAI to their respective roles using shorter, less dense sentences. The revised abstract has been updated in the manuscript accordingly.
>
> Specifically, we now first explain that VLMs suffer from insufficient coordination of cross-attention across layers and patches, leading to attention drift and suboptimal convergence, before introducing CCRA. We then clearly state that LPWCA is designed to capture fine-grained region–semantic correlations and reweight layer/patch embeddings, while PAI progressively refines attention from global semantics to local regions to stabilize alignment.
>
> Thank you again for your patience and invaluable feedback.
>
> Best regards,
>
> The Authors

---

### Author Response · Authors · 2025-11-30
**Summary of Contributions and Rebuttal Updates for AC**

Dear Area Chair,

We understand the heavy workload involved in the decision-making process. To facilitate your assessment, we provide a concise summary of our rebuttal, starting with our core contribution to the ICLR community.

# 1. Value to the Community: Efficient & Consistent Vision-Language Alignment

 **(1) Motivation:**
We identify the critical bottleneck of "Attention Drift" in current multimodal alignment. This occurs when layer-wise and patch-wise cross-attention assign incompatible regions to the same textual query. Then, it leads to unstable convergence and suboptimal model performance.

**(2) Solution:**
We propose CCRA, a lightweight alignment module that resolves this drift via "Consistency-driven Progressive Attention Coordination". Unlike simple stacking, we enforce a strict logical hierarchy: Global Alignment (LPWCA) $\rightarrow$ Semantic Refinement (LWCA) $\rightarrow$ Local Grounding (PWCA), ensuring visual features are internally consistent with the text query before entering the LLM.

**(3) Impact:**
We demonstrate that better alignment, not just larger models, can unlock significant gains. With negligible cost (3.55M parameters), CCRA enables a 7B VLM model to outperform a 13B VLM baseline. This provides the ICLR community with a foundational perspective on building stable, interpretable, and efficient alignment module.

# 2. Clarification on Novelty: Consistency-driven Progressive Coordination
Some reviewers (z7Cf, XkdG) argue that our method is merely a simple combination of existing modules. However, we clarified that CCRA is not a mere stacking of blocks, but a theoretically motivated "Consistency-driven Progressive Attention Coordination" framework. Also, it supported by comprehensive experimental evidence.

**(1) Logical Hierarchy:**
We adhere to a strict dependency: Global Alignment $\rightarrow$ Semantic Refinement $\rightarrow$ Local Grounding. This mirrors human cognitive patterns and is mathematically formulated to minimize gradient conflict. (See our reply to Reviewer HzMb 's Question2-1)

**(2) Empirical Evidence:**
In our rebuttal (Appendix C.3, Fig. 8), we added training loss analyses. The results prove that our specific progressive order yields smooth, stable convergence, whereas decoupled or shuffled orders lead to oscillation.  This corroborates Figure 6 in our main text, which visually demonstrates that our strategy produces significantly sharper, drift-free attention maps compared to decoupled or shuffled baselines.

**(3) Regarding Reviewer z7Cf:**
We highlight that he explicitly admitted the work "deserves to be published" and their "concerns are well addressed," yet maintained a negative rating solely due to a subjective view on "depth." We believe our effective solution to "Attention Drift" and proven generalization clearly establishes the significance required for ICLR.

# 3. Robustness in Significance and Generalization
We addressed concerns of reviewers (z7Cf, bDBa) regarding the baselines and inference overhead with extensive new experiments:

**(1) Outperforming Larger Models with Minimal Cost:**
CCRA adds only 3.55M parameters. With this negligible cost, our 7B model outperforms the LLaVA-v1.5-13B baseline across 11 benchmarks.

**(2) Generalization to SOTA Architectures:**
We extended our experiments to LLaVA-NeXT (Llama-3-8B), LLaVA-v1.5-13B, MiniGPT-v2, and LLaVA-Adapter. CCRA consistently yields improvements (e.g., +1.4~2.2 avg gains) with different Visual Encoders and LLM Backbones (See our reply to Reviewer bDBa's Question2-1)

**(3) Negligible Latency:**
We provided a breakdown showing CCRA adds only 6.5ms to the prefill time, with zero impact on token generation speed (tokens/s). (Please refer to our detailed responses to Reviewer z7Cf and bDBa)

In closing, we thank the reviewers for their constructive feedback, which has significantly strengthened our manuscript. We represent a solid step towards more efficient and interpretable vision-language alignment and sincerely hope this work can contribute to the ICLR community.

Best regards,
The Authors

---

### Note · Program_Chairs · 2026-01-17
**Submission Desk Rejected by Program Chairs**

The following references in this submission do not refer to real documents and/or have major errors in bibliographic information:

 Zhaoyang Zhu et al. Seed-bench: Benchmarking multimodal language models with generative completions. arXiv preprint arXiv:2307.16102, 2023.
Yuzhe Yao et al. Pope: Probing lmms' visual grounding via perception-oriented probing evaluation. arXiv preprint arXiv:2309.09507, 2023.